# OPEN-WORLD REINFORCEMENT LEARNING OVER LONG SHORT-TERM IMAGINATION

**Jiajian Li**[1*]   **Qi Wang**[1,2*]   **Yunbo Wang**[1†]   **Xin Jin**[2]   **Yang Li**[3]   **Wenjun Zeng**[2]   **Xiaokang Yang**[1]

[1] MoE Key Lab of Artificial Intelligence, AI Institute, Shanghai Jiao Tong University, Shanghai, China
[2] Ningbo Institute of Digital Twin, Eastern Institute of Technology, Ningbo, China
[3] School of Computer Science and Technology, East China Normal University, Shanghai, China
https://qiwang067.github.io/ls-imagine

## ABSTRACT

Training visual reinforcement learning agents in a high-dimensional open world presents significant challenges. While various model-based methods have improved sample efficiency by learning interactive world models, these agents tend to be "short-sighted", as they are typically trained on short snippets of imagined experiences. We argue that the primary challenge in open-world decision-making is improving the exploration efficiency across a vast state space, especially for tasks that demand consideration of long-horizon payoffs. In this paper, we present LS-Imagine, which extends the imagination horizon within a limited number of state transition steps, enabling the agent to explore behaviors that potentially lead to promising long-term feedback. The foundation of our approach is to build a *long short-term world model*. To achieve this, we simulate goal-conditioned jumpy state transitions and compute corresponding affordance maps by zooming in on specific areas within single images. This facilitates the integration of direct long-term values into behavior learning. Our method demonstrates significant improvements over state-of-the-art techniques in MineDojo.

## 1   INTRODUCTION

Open-world decision-making in the context of reinforcement learning (RL) involves the following characteristics: (i) The agent operates within an interactive environment that features a vast state space; (ii) The learned policy presents a high degree of flexibility, allowing interaction with various objects in the environment; (iii) The agent lacks full visibility of the internal states and physical dynamics of the external world, meaning that its perception of the environment (*e.g.*, raw images) carries substantial uncertainty. For example, Minecraft serves as a typical open-world game.

Building upon recent progress in visual control, open-world decision-making aims to train agents to approach human-level intelligence based solely on high-dimensional visual observations. However, this presents significant challenges. For example, in Minecraft tasks, existing methods like Voyager (Wang et al., 2024a) employ specific Minecraft APIs as the high-level controller, which is incompatible with standard visual control settings. While approaches such as PPO-with-MineCLIP (Fan et al., 2022) and DECKARD (Nottingham et al., 2023) perform low-level visual control, these model-free RL methods struggle to grasp the underlying mechanics of the environment. This may result in high trial-and-error costs, leading to inefficiencies in both exploration and sample usage. Although DreamerV3 (Hafner et al., 2023) employs a model-based RL (MBRL) approach to improve sample efficiency, it is often "short-sighted" since the policy is optimized using short-term experiences—typically 15 time steps—generated by the world model. The absence of long-term guidance significantly hampers an effective exploration of the vast solution space of the open world.

To improve the behavior learning efficiency of MBRL, in this paper, we introduce a novel method named Long Short-Term Imagination (LS-Imagine). Our key approach involves *enabling the world model to efficiently simulate the long-term effects of specific behaviors without the need for repeatedly*

---

*Equal contribution.
†Corresponding author: Yunbo Wang <yunbow@sjtu.edu.cn>.

Figure 1: The general framework of LS-Imagine, an MBRL agent that operates solely on raw pixels. The fundamental idea is to extend the imagination horizon within a limited number of state transition steps, enabling the agent to explore behaviors that potentially lead to promising long-term feedback.

*rolling out one-step predictions.* As illustrated in Figure 1, once trained, the world model provides both instant and jumpy state transitions[1] along with corresponding (intrinsic) rewards, facilitating policy optimization in a joint space of short- and long-term imaginations. This encourages the agent to explore behaviors that lead to promising long-term outcomes.

The foundation of LS-Imagine is to train a *long short-term world model*, which requires integrating task-specific guidance into the representation learning phase based on off-policy experience replay. However, this creates a classic "chicken-and-egg" dilemma: *without true data showing the agent has reached the goal, how can we effectively train the model to simulate jumpy transitions from current states to pivotal future states that suggest a high likelihood of achieving that goal?* To address this issue, we first continuously zoom in on individual images to simulate the consecutive video frames as the agent approaches the goal. We then generate affordance maps[2] by evaluating the relevance of the pseudo video to task-specific goals presented in textual instructions, using the established MineCLIP reward model (Fan et al., 2022). Subsequently, we train specific branches of the world model to capture both instant and jumpy state transitions, using pairs of image observations from adjacent time steps as well as those across longer intervals. Finally, we optimize the agent's policy based on a finite sequence of imagined latent states generated by the world model, integrating a more direct estimate of long-term values into decision-making.

Let's use the example in Figure 1 to further elaborate the novel aspects of the behavior learning process: After receiving the instruction "cut a tree", the agent simulates near-future states based on the current real observation. It initially performs several single-step rollouts until it identifies a point in time for a long-distance state jump that allows it to approach the tree. The agent then executes this jump and optimizes its policy network to maximize the long-sight value function.

We evaluate our approach in the challenging open-world tasks from MineDojo (Fan et al., 2022). LS-Imagine demonstrates superior performance compared to existing visual RL methods.

The contributions of this work are summarized as follows:

- We present a novel model-based RL method that captures both instant and jumpy state transitions and leverages them in behavior learning to improve exploration efficiency in the open world.
- Our approach presents four concrete contributions: (i) a long short-term world model architecture, (ii) a method for generating affordance maps through image zoom-in, (iii) a novel form of intrinsic rewards based on the affordance map, and (iv) an improved behavior learning method that integrates long-term values and operates on a mixed long short-term imagination pathway.

## 2 PROBLEM FORMULATION AND NOTATIONS

We solve visual reinforcement learning as a partially observable Markov decision process (POMDP), using MineDojo as the test bench. Specifically, our method manipulates low-dimensional control signals $a_t$ while receiving only sequential high-dimensional visual observations $o_{<t}$ and episodic sparse rewards $r^{\text{env}}$, without access to the internal APIs of the open-world games. In comparison, as

---

[1]As shown in Figure 1, a jumpy transition allows the agent to bypass intermediate states and directly simulate a task-relevant future state $s_{t+H}$ in one step. This process occurs exclusively during world model imagination.

[2]Affordance maps highlight regions within an observation that are potentially relevant to the task (Qi et al., 2020; Wang et al., 2022).

Table 1: Experimental setups of the Minecraft AI agents. *IL* is short for imitation learning.

| Model | Controller | Observation | Video Demos |
|---|---|---|---|
| DECKARD (2023) | RL | Pixels & Inventory | ✓ |
| Auto MC-Reward (2024a) | IL + RL | Pixels & GPS | ✗ |
| Voyager (2024a) | GPT-4 | Minecraft simulation & Error trace | ✗ |
| DEPS (2023) | IL | Pixels & Yaw/pitch angle & GPS & Voxel | ✗ |
| STEVE-1 (2023) | Generative model | Pixels | ✗ |
| VPT (2022) | IL + RL | Pixels | ✓ |
| DreamerV3 (2023) | RL | Pixels | ✗ |
| LS-Imagine | RL | Pixels | ✗ |

shown in Table 1, existing Minecraft agents present notable distinctions in learning paradigms (*i.e.*, controller), observation data, and the use of expert demonstrations.

The world model presented in this paper consists of two main components: a short-term transition branch and a long-term imagination branch. As a result, it employs a complex notation system. We now introduce the key notations that will be frequently used throughout the paper:

- $\mathcal{M}_t$ represents the affordance map.
- $c_t$ denotes the episode continuation flag.
- $j_t$ is the jumping flag that triggers jumpy state imaginations.
- $\Delta_t$ represents the number of environmental steps between the jumpy transitions.
- $G_t$ is the cumulative reward over $\Delta_t$.

We use $(o'_t, a'_t, \mathcal{M}'_t, r'_t, c'_t, j'_t, \Delta'_t, G'_t)$ to represent the simulated environment data that are used to train the long-term imagination branch of the world model. The policy is learned on trajectories of mixed long- and short-term imaginations $\{(\hat{s}_t, \hat{a}_t, \hat{r}_t, \hat{c}_t, \hat{j}_t, \hat{\Delta}_t, \hat{G}_t)\}$, where $\hat{s}_t$ represents the latent state, and the variables predicted by the model are indicated using the superscript ($\hat{\ }$).

## 3 METHOD

### 3.1 OVERVIEW OF LS-IMAGINE

In this section, we present the details of LS-Imagine, which involves the following algorithm steps, including world model learning, behavior learning, and environment interaction:

1. *Affordance map computation* (Sec. 3.2.1): We employ a sliding bounding box to scan individual images and execute continuous zoom-ins inside the bounding box, simulating consecutive video frames that correspond to long-distance state transitions. We then create affordance maps by assessing the relevance of the fake video clips to task-specific goals expressed in text using the established MineCLIP reward model (Fan et al., 2022).

2. *Rapid affordance map generation* (Sec. 3.2.2): Given that affordance maps will be frequently used in subsequent Step 5 to evaluate the necessities for jumpy state transitions, we train a U-Net module to approximate the affordance maps annotated in Step 1 for the sake of efficiency.

3. *World model training* (Sec. 3.3): We train the world model to capture short- and long-term state transitions, using replay data with high responses from the affordance map. Each trajectory from the buffer includes pairs of samples from both adjacent time steps and long-distance intervals.

4. *Behavior learning* (Sec. 3.4): We perform an actor-critic algorithm to optimize the agent's policy based on a finite sequence of long short-term imaginations generated by the world model.

5. *Data update*: We apply the agent to interact with the environment and gather new data. Next, we leverage the generated affordance map to efficiently filter sample pairs suitable for long-term modeling, incorporating both short- and long-term sample pairs to update the replay buffer.

6. Iterate Steps 3–5.

Below, we discuss each training step in detail. The full algorithm can be found in Appendix C.4.

### 3.2 AFFORDANCE MAP AND INTRINSIC REWARD

We generate affordance maps using visual observations and textual task definitions to improve the sample efficiency of model-based reinforcement learning in open-world tasks. The core idea is to direct the agent's attention to task-relevant areas of the visual observation, leading to higher exploration

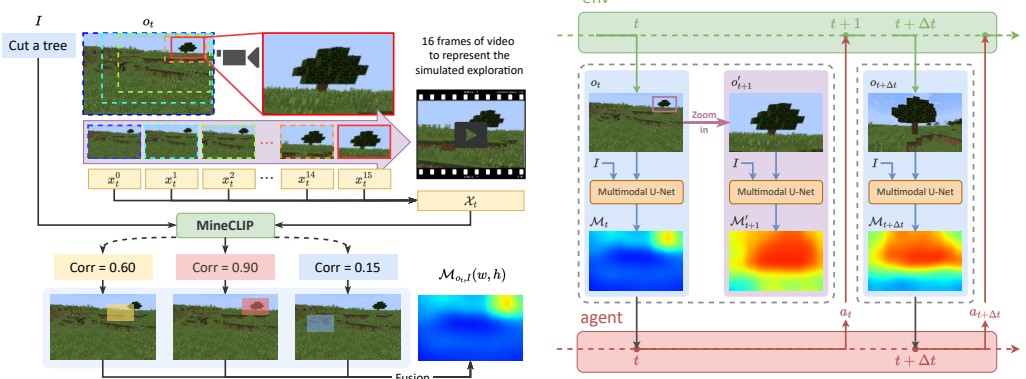

(a) Affordance map calculation        (b) Rapid affordance map generation

Figure 2: The two steps for on-the-fly affordance map estimation: (a) Simulate exploration via image zoom-in and calculate the task-correlation scores of the virtual explorations using MineCLIP. (b) Learn to generate affordance maps more efficiently using a multimodal U-Net.

efficiency. Let $\mathcal{M}_{o_t,I}(w,h)$ be the affordance map that represents the potential exploration value at pixel position $(w,h)$ on the image observation $o_t$, given textual instruction $I$ (*e.g.*, "cut a tree"). The affordance map highlights the relevance between regions of the observation and the task description, serving as a spatial prior that effectively directs the agent's exploration toward areas of interest.

### 3.2.1 Affordance Map Computation via Virtual Exploration

To create the affordance map, as shown in Figure 2(a), we simulate and evaluate the agent's exploration without relying on real successful trajectories. Concretely, we first adopt a random agent to interact with task-specific environments for data collection. Starting with the agent's observation $o_t$ at time step $t$, we use a sliding bounding box with dimensions scaled to $15\%$ of the observation's width and height to traverse the entire observation from left to right and top to bottom. The sliding bounding box moves horizontally and vertically in 9 steps, respectively, covering every potential region in both dimensions. For each position on the sliding bounding box of the observation $o_t$, we crop 16 images from $o_t$. These cropped images narrow the field of view to focus on the region and are resized back to the original image dimensions. These resized images are denoted as $x_t^k$ (where $0 \leq k < 16$). The ordered set $\mathcal{X}_t = [x_t^k \mid k = 0, 1, \ldots, 15]$ represents a sequence of 16 frames simulating the visual transition as the agent moves towards the position specified by the current sliding bounding box. Subsequently, we employ the MineCLIP model[3] to calculate the correlation between the $\mathcal{X}_t$ of images, simulating the virtual exploration process, and the task description $I$. In this way, we quantify the affordance value of the sliding bounding box, indicating the potential exploration interest of the area. After calculating the correlation score for each sliding bounding box, we fuse these values to obtain a smooth affordance map $\mathcal{M}_{o_t,I}$. For pixels that are covered by multiple sliding bounding boxes due to overlapping regions, the integrated affordance value is obtained by averaging the values from all the overlapping windows.

### 3.2.2 Multimodal U-Net for Rapid Affordance Map Generation

The annotation of affordance maps, as previously described, involves extensive window traversal and computations for each window position using a pre-trained video-text alignment model. This method is computationally demanding and time-consuming, making real-time applications challenging. To address this issue, we first use a random agent to interact with the environment for data collection. Next, we annotate the affordance maps for the collected images using the aforementioned method based on virtual exploration. We gather a dataset of tuples $(o_t, I, \mathcal{M}_{o_t,I})$ and use it to train a multimodal U-Net based on Swin-Unet (Cao et al., 2022). To handle multimodal inputs, we extract text features from the language instructions and image features from the downsampling process of Swin-Unet, and fuse them with multi-head attention. We present architecture details in Figure 9 in the appendix. In this way, with the pretrained multimodal U-Net, we can efficiently generate affordance maps at each time step using visual observations and language instructions.

---

[3]MineCLIP (Fan et al., 2022) pretrains a video-language representation using Minecraft videos, enabling it to compute the correlation between a text string and a 16-frame video segment.

### 3.2.3 AFFORDANCE-DRIVEN INTRINSIC REWARD

To leverage the task-relevant prior knowledge presented by the affordance map for efficient exploration in the open world, we introduce the following intrinsic reward function:

$$r_t^{\text{intr}} = \frac{1}{WH} \sum_{w=1}^{W} \sum_{h=1}^{H} \mathcal{M}_{o_t,I}(w,h) \cdot \mathcal{G}(w,h), \tag{1}$$

where $W$ and $H$ denote the width and height of the visual observation. $\mathcal{G}$ represents a Gaussian matrix with dimensions matching those of the affordance map. It corresponds to a 2D Gaussian distribution, with its peak located at the center of the affordance map. The values in the matrix are determined by standard deviations $(\sigma_x, \sigma_y)$, while the mean is uniformly set to 1 across the entire matrix. We present visualizations of $\mathcal{G}$ and conduct hyperparameter analyses on $(\sigma_x, \sigma_y)$ in Appendix D.6. The intuition behind this design is to encourage the agent to move toward the target.

Overall, the agent receives a composite reward consisting of the episodic sparse reward from the environment, the reward from MineCLIP (Fan et al., 2022), and the intrinsic reward from the affordance map: $r_t = r_t^{\text{env}} + r_t^{\text{MineCLIP}} + \alpha r_t^{\text{intr}}$, where $\alpha$ is a hyperparameter. In contrast to the MineCLIP reward, which relies on the agent's past performance, our affordance-driven intrinsic reward emphasizes long-term values derived from future virtual exploration. It encourages the agent to adjust the policy to pursue task-related targets when they appear in its view, ensuring these targets are centrally positioned in future visual observations to maximize this reward function.

## 3.3 LONG SHORT-TERM WORLD MODEL

### 3.3.1 LEARNING JUMPING FLAGS

In LS-Imagine, the world model is customized for long-term and short-term state transitions. It decides which type of transition to adopt based on the current state and predicts the next state with the selected transition branch. To facilitate the switch between long-term and short-term state transitions, we introduce a jumping flag $j_t$, which indicates whether a jumpy transition or long-term state transition, should be adopted at time step $t$. When a distant task-related target appears in the agent's observation, which can be reflected by a higher kurtosis in the affordance map, a jumpy transition allows the agent to imagine the future state of approaching the target. To this end, we define relative kurtosis $K_r$ which measures whether there are significantly higher target areas than the surrounding areas in the affordance map, and absolute kurtosis $K_a$ represents the confidence level of target presence in that area. Formally,

$$K_r = \frac{1}{WH} \sum_{w=1}^{W} \sum_{h=1}^{H} \left[ \left( \frac{\mathcal{M}_{o,I}(w,h) - \text{mean}(\mathcal{M}_{o,I})}{\text{std}(\mathcal{M}_{o,I})} \right)^4 \right],$$
$$K_a = \max(\mathcal{M}_{o,I}) - \text{mean}(\mathcal{M}_{o,I}). \tag{2}$$

To normalize the relative kurtosis, we apply the sigmoid function to it, and then multiply it by the absolute kurtosis to calculate the jumping probability:

$$P_{\text{jump}} = \text{sigmoid}(K_r) \times K_a. \tag{3}$$

The jumping probability measures the confidence in the presence of task-relevant targets far from the agent in the visual observation. To determine whether to employ long-term state transition, we use a dynamic threshold, which is the mean of the collected jumping probabilities at each time step, plus one standard deviation. For a detailed explanation, please refer to C.1. If $P_{\text{jump}}$ exceeds this threshold, the jump flag $j_t$ is True and the agent switches to jumpy state transitions in the imagination phase.

### 3.3.2 LEARNING JUMPY STATE TRANSITIONS

In LS-Imagine, the state transition model includes both short-term and long-term branches. As shown in Figure 3 (a), the short-term transition model integrates the previous deterministic recurrent state $h_{t-1}$, stochastic state $z_{t-1}$, and action $a_{t-1}$ to adopt the single-step transition. In contrast, the long-term branch simulates jumpy state transitions toward the target. It is important to clarify that the index $t$ does not denote the time step during real environmental interactions but instead represents the positional order of states within the imagination sequence. The overall world model of LS-Imagine is

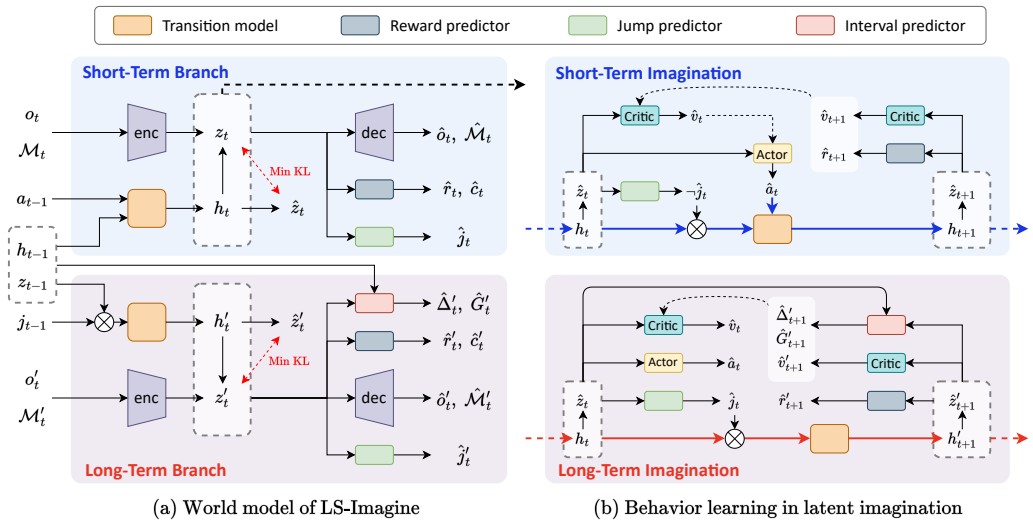

(a) World model of LS-Imagine      (b) Behavior learning in latent imagination

Figure 3: The overall architecture of the world model and the behavior learning process.

primarily based on DreamerV3 (Hafner et al., 2023), with novel components specifically designed to capture jumpy state transitions:

$$
\begin{array}{ll}
\text{Short-term transition model:} & h_t = f_\phi(h_{t-1}, z_{t-1}, a_{t-1}) \\[4pt]
\text{Long-term transition model:} & h'_t = f_\phi(h_{t-1}, z_{t-1}) \\[4pt]
\text{Encoder:} & z_t \sim q_\phi(z_t \mid h_t, o_t, \mathcal{M}_t) \\[4pt]
\text{Dynamics predictor:} & \hat{z}_t \sim p_\phi(\hat{z}_t \mid h_t) \\[4pt]
\text{Reward predictor:} & \hat{r}_t, \hat{c}_t \sim p_\phi(\hat{r}_t, \hat{c}_t \mid h_t, z_t) \\[4pt]
\text{Decoder:} & \hat{o}_t, \hat{\mathcal{M}}_t \sim p_\phi(\hat{o}_t, \hat{\mathcal{M}}_t \mid h_t, z_t) \\[4pt]
\text{Jump predictor:} & \hat{j}_t \sim p_\phi(\hat{j}_t \mid h_t, z_t) \\[4pt]
\text{Interval predictor:} & \hat{\Delta}'_t, \hat{G}'_t \sim p_\phi(\hat{\Delta}'_t, \hat{G}'_t \mid h_{t-1}, z_{t-1}, h'_t, z'_t)
\end{array}
\tag{4}
$$

At time step $t$, we feed the recurrent state $h_t$, the observation $o_t$, and the affordance map $\mathcal{M}_t$ into the encoder to obtain posterior state $z_t$. We also use the affordance map as an input of the encoder, which serves as the goal-conditioned prior guidance to the agent. Notably, the prediction of prior state $\hat{z}_t$ does not involve the current observation or affordance map, relying solely on historical information. We use $(h_t, z_t)$ to reconstruct the visual observation $\hat{o}_t$ and the affordance map $\hat{\mathcal{M}}_t$, and predict the reward $\hat{r}_t$, episode continuation flag $\hat{c}_t$, and jumping flag $\hat{j}_t$. For long-term state transitions, we use an interval predictor to estimate the expected number of interaction steps $\hat{\Delta}'_t$ required to transition from the pre-jump state $(h_{t-1}, z_{t-1})$ to the post-jump state $(h'_t, z'_t)$, along with the expected cumulative reward $\hat{G}'_t$ that the agent may receive during this time interval. We detail the approach to annotate $\Delta'_t$ and $G'_t$ using the real interaction data in Appendix C.1.

We collect short-term tuples $\mathcal{D}_t = (o_t, a_t, \mathcal{M}_t, r_t, c_t, j_t, \Delta_t, G_t)$ from each interaction with the environment using the current policy. When observing $j_t = 1$, we additionally construct long-term tuples $\mathcal{D}'_{t+1} = (o'_{t+1}, a'_{t+1}, \mathcal{M}'_{t+1}, r'_{t+1}, c'_{t+1}, j'_{t+1}, \Delta'_{t+1}, G'_{t+1})$ based on $\mathcal{D}_t$. More details for this process can be found in Appendix C.1. During representation learning, we sample short-term tuples $\{\mathcal{D}_t\}_{t=1}^T$ and the long-term tuples following jumpy transitions $\{\mathcal{D}'_{t+1}\}_{t \in \mathcal{T}}$ from the replay buffer $\mathcal{B}$, where $\mathcal{T}$ denotes the set of time steps at which long-term state transitions are required. The loss functions for each component of the short-term and long-term world model branch are as follows:

$$
\text{Short-term branch:}
\begin{cases}
\mathcal{L}_{\text{dyn}} \doteq \max\left(1, \text{KL}\left[\text{sg}\left(q_\phi\left(z_t \mid h_t, o_t, \mathcal{M}_t\right)\right) \parallel p_\phi\left(z_t \mid h_t\right)\right]\right) \\[4pt]
\mathcal{L}_{\text{enc}} \doteq \max\left(1, \text{KL}\left[q_\phi\left(z_t \mid h_t, o_t, \mathcal{M}_t\right) \parallel \text{sg}\left(p_\phi\left(z_t \mid h_t\right)\right)\right]\right) \\[4pt]
\mathcal{L}_{\text{dec}} \doteq -\ln p_\phi\left(o_t, \mathcal{M}_t \mid h_t, z_t\right) \\[4pt]
\mathcal{L}_{\text{pred}} \doteq -\ln p_\phi\left(r_t, c_t \mid h_t, z_t\right) - \ln p_\phi\left(j_t \mid h_t, z_t\right)
\end{cases}
\tag{5}
$$

$$\text{Long-term branch:} \quad \begin{cases} \mathcal{L}'_{\text{dyn}} \doteq \max\left(1, \text{KL}\left[\text{sg}\left(q_\phi\left(z'_t \mid h'_t, o'_t, \mathcal{M}'_t\right)\right) \parallel p_\phi\left(z'_t \mid h'_t\right)\right]\right) \\ \mathcal{L}'_{\text{enc}} \doteq \max\left(1, \text{KL}\left[q_\phi\left(z'_t \mid h'_t, o'_t, \mathcal{M}'_t\right) \parallel \text{sg}\left(p_\phi\left(z'_t \mid h'_t\right)\right)\right]\right) \\ \mathcal{L}'_{\text{dec}} \doteq -\ln p_\phi\left(o'_t, \mathcal{M}'_t \mid h'_t, z'_t\right) \\ \mathcal{L}'_{\text{pred}} \doteq -\ln p_\phi(r'_t, c'_t \mid h'_t, z'_t) - \ln p_\phi(j'_t \mid h'_t, z'_t) \\ \mathcal{L}'_{\text{int}} \doteq -\ln p_\phi\left(\Delta'_t, G'_t \mid h_{t-1}, z_{t-1}, h'_t, z'_t\right) \end{cases} \quad . \quad (6)$$

We can optimize the world model $\mathcal{W}_\phi$ by minimizing over replay buffer $\mathcal{B}$:

$$\begin{aligned} \mathcal{L} \doteq \mathbb{E}\Big[\sum\nolimits_{\{\mathcal{D}_t\}_{t=1}^T} \left(\beta_{\text{dyn}}\mathcal{L}_{\text{dyn}} + \beta_{\text{enc}}\mathcal{L}_{\text{enc}} + \beta_{\text{pred}}\left(\mathcal{L}_{\text{dec}} + \mathcal{L}_{\text{pred}}\right)\right) + \\ \beta_{\text{long}}\sum\nolimits_{\{\mathcal{D}'_{t+1}\}_{t\in\mathcal{T}}} \left(\beta_{\text{dyn}}\mathcal{L}'_{\text{dyn}} + \beta_{\text{enc}}\mathcal{L}'_{\text{enc}} + \beta_{\text{pred}}\left(\mathcal{L}'_{\text{dec}} + \mathcal{L}'_{\text{pred}} + \mathcal{L}'_{\text{int}}\right)\right)\Big]. \end{aligned} \quad (7)$$

## 3.4 BEHAVIOR LEARNING OVER MIXED LONG SHORT-TERM IMAGINATIONS

As shown in Figure 3 (b), LS-Imagine employs an actor-critic algorithm to learn behavior from the latent state sequences predicted by the world model. The goal of the actor is to optimize the policy to maximize the discounted cumulative reward $R_t$, while the role of the critic is to estimate the discounted cumulative rewards using the current policy for each state $\hat{s}_t \doteq \{h_t, \hat{z}_t\}$:

$$\text{Actor:} \quad \hat{a}_t \sim \pi_\theta\left(\hat{a}_t \mid \hat{s}_t\right), \quad \text{Critic:} \quad v_\psi\left(\hat{R}_t \mid \hat{s}_t\right). \quad (8)$$

Starting from the initial state encoded from the sampled observation and the affordance map, we dynamically select either the long-term transition model or the short-term transition model to predict subsequent states based on the jumping flag $\hat{j}_t$. For the long short-term imagination sequence $\{(\hat{s}_t, \hat{a}_t)\}_{t=1}^L$ with an imagination horizon of $L$, we predict reward sequence $\hat{r}_{1:L}$ and the continuation flag sequence $\hat{c}_{1:L}$ through the reward predictor. Similar to Eq. (4), the index $t$ does not represent the time step in the environment, but rather the positional order of the states in the imagination sequence. Specifically, starting from state $\hat{s}_t$, any subsequent state obtained via either a short-term transition or a long-term transition is indexed sequentially as $t + 1$.

For jumpy states predicted by long-term imagination, the interval predictor estimates (i) the number of steps $\hat{\Delta}_t$ from $\hat{s}_{t-1}$ to $\hat{s}_t$ and (ii) the potential discounted cumulative reward $\hat{G}_t$ over the time interval of $\hat{\Delta}_t$. Otherwise, for states obtained via short-term imagination, which correspond to single-step transitions in the environment, we set $\hat{\Delta}_t = 1$ and $\hat{G}_t = \hat{r}_t$. Consequently, within one imagination episode, we obtain a sequence of step intervals $\hat{\Delta}_{2:L}$ and a sequence of predicted rewards $\hat{G}_{2:L}$ between consecutive imagination states.

We employ a modified bootstrapped $\lambda$-returns that considers both long-term and short-term imaginations to calculate the discounted cumulative rewards for each state:

$$R_t^\lambda \doteq \begin{cases} \hat{c}_t\{\hat{G}_{t+1} + \gamma^{\hat{\Delta}_{t+1}}\left[(1-\lambda)v_\psi(\hat{s}_{t+1}) + \lambda R_{t+1}^\lambda\right]\} & \text{if } t < L \\ v_\psi(\hat{s}_L) & \text{if } t = L \end{cases} \quad . \quad (9)$$

The critic uses the maximum likelihood loss to predict the distribution of the return estimates $R_t^\lambda$:

$$\mathcal{L}(\psi) \doteq -\sum_{t=1}^L \ln p_\psi\left(R_t^\lambda \mid \hat{s}_t\right). \quad (10)$$

Following DreamerV3 (Hafner et al., 2023), we train the actor to maximize the return estimates $R_t^\lambda$. Notably, since long-term imagination does *not* involve actions, we do not optimize the actor at time steps when jumpy state transitions are adopted. Therefore, unlike DreamerV3, we apply an additional factor of $(1 - \hat{j}_t)$ to ignore updates at long-term imagination steps:

$$\mathcal{L}(\theta) \doteq -\sum_{t=1}^L \text{sg}\left[\left(1 - \hat{j}_t\right)\frac{R_t^\lambda - v_\psi(\hat{s}_t)}{\max(1, S)}\right]\log \pi_\theta(\hat{a}_t \mid \hat{s}_t) + \eta\, \text{H}\left[\pi_\theta(\hat{a}_t \mid \hat{s}_t)\right]. \quad (11)$$

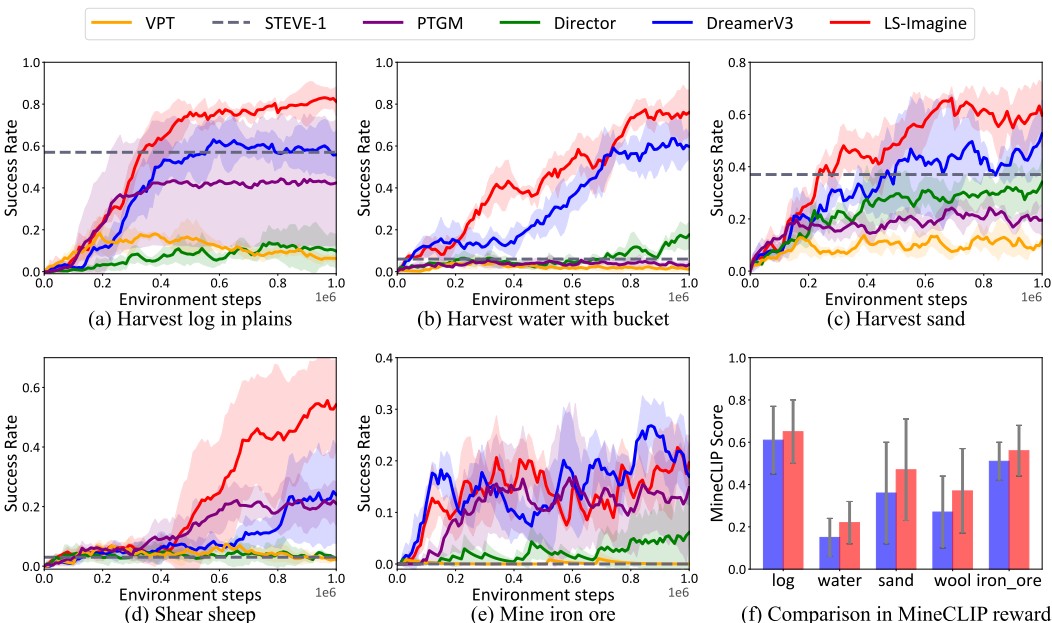

Figure 4: Comparison of LS-Imagine against strong Minecraft agents, including *DreamerV3* (Hafner et al., 2023), *VPT* (Baker et al., 2022), *STEVE-1* (Lifshitz et al., 2023), *PTGM* (Yuan et al., 2024), and *Director* (Hafner et al., 2022). We present the numerical results in Table 3 in the appendix.

## 4 EXPERIMENTS

We explore LS-Imagine on the challenging MineDojo (Fan et al., 2022) benchmark on top of the popular Minecraft game, which is a comprehensive simulation platform with various open-ended tasks. We use 5 tasks, *i.e., harvest log in plains*, *harvest water with bucket*, *harvest sand*, *shear sheep*, and *mine iron ore*. These tasks demand numerous steps to complete and present significant challenges for agent learning. We adopt a binary reward that indicates whether the task was completed, along with the MineCLIP reward (Fan et al., 2022). Further details of the environmental setups are provided in Appendix A. Besides, we introduce the compared models in Appendix B.

**Implementation details.** We conduct our experiments on the MineDojo environment, where both visual observation and corresponding affordance maps are resized to $64 \times 64$ pixels. To generate accurate affordance maps, we collect 2,000 images from the environment using a random agent under the current task instruction and generate a discrete set of $(o_t, I, \mathcal{M}_{o_t, I})$, which are then used to finetune the multimodal U-Net for 200 epochs. For tasks in the MineDojo benchmark, we train the agent for $1 \times 10^6$ environment steps. Each training of LS-Imagine takes approximately 23 GB of VRAM and requires around 1.7 days to complete on a single RTX 3090 GPU.

### 4.1 MAIN COMPARISON

We evaluate all the Minecraft agents in terms of success rate shown in Figure 4 and per-episode steps shown in Figure 5. We find that LS-Imagine significantly outperforms the compared models, particularly in scenarios where sparse targets are distributed in the task. In Figure 4 (f), we showcase the MineCLIP values achieved by LS-Imagine and DreamerV3. Specifically, a sliding window of length 16 is used to compute the local MineCLIP values for each segment. The mean value is then calculated from all sliding windows. We can see that agents trained using our method achieve higher MineCLIP values within a single episode compared to DreamerV3. This suggests that LS-Imagine facilitates quicker detection of task-relevant visual targets in open-world environments.

Additionally, we present qualitative results in Figure 6(a). In the top row, we decode the latent states before and after the jumpy state transitions back to the pixel space. To better understand how affordance maps facilitate the jumpy state transitions and whether they can provide effective goal-conditioned guidance, the bottom rows visualize the affordance maps reconstructed from the latent states. These visualizations demonstrate that the proposed world model can adaptively determine when to utilize long-term imagination based on the current visual observation. Furthermore, the

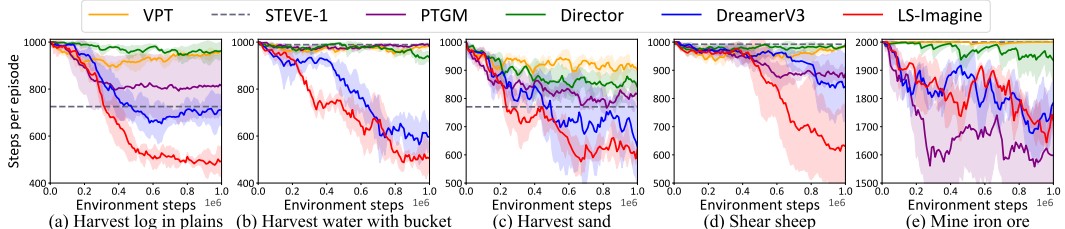

Figure 5: The number of steps per episode for task completion.

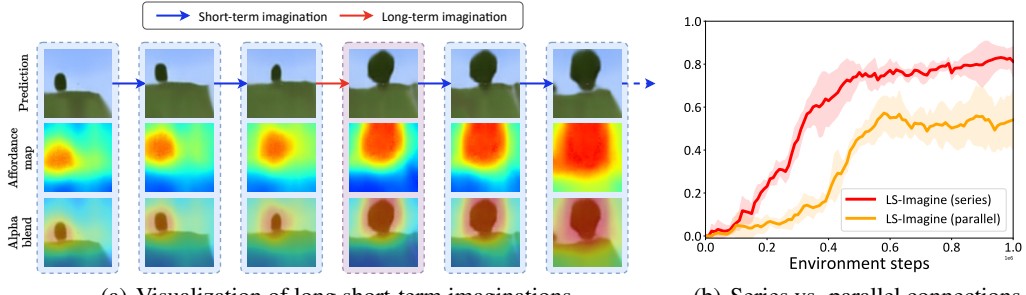

(a) Visualization of long short-term imaginations

(b) Series vs. parallel connections

Figure 6: (a) Visualization of long short-term imaginations and (b) a further discussion on possible architecture designs of Series and Parallel connections of these two imagination pathways.

generated affordance maps align effectively with areas that are highly relevant to the final goal, thereby enabling the agent to perform more efficient policy exploration.

## 4.2 MODEL ANALYSES

**Ablation studies.** We conduct the ablation studies to validate the effect of the affordance-driven intrinsic reward and long short-term imagination. Figure 7 presents corresponding results in the challenging MineDojo tasks. As shown by the blue curve, removing the long-term imagination of LS-Imagine leads to a performance decline, which indicates the necessity of introducing long-term imagination and switching between it and short-term imagination adaptively. For the model represented by the green curve, we do not employ affordance-driven intrinsic reward. It shows that the affordance-driven intrinsic reward also plays an important role during the early training stage of agents. Additionally, unlike the MineCLIP reward being calculated based on a series of

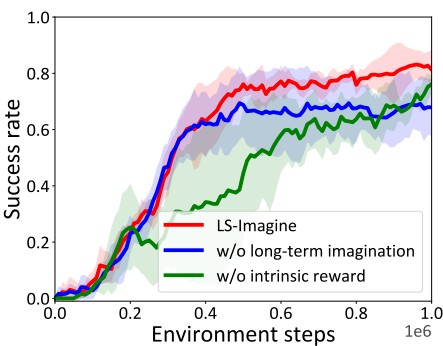

Figure 7: Ablation study results.

states, the affordance-driven intrinsic reward relies solely on a single independent state. This approach enables a more accurate estimation of the reward for the post-jumpy-transition state.

**Alternative pathways of mixed imaginations.** It is worth highlighting that the long short-term imagination is implemented sequentially. In Figure 10(a) in the appendix, we provide a visualization illustrating how the agent sequentially performs short-term and long-term imaginations within a single sequence. Alternatively, as illustrated in Figure 10(b), we could structure long- and short-term imagination pathways in parallel. Specifically, we begin by applying short-term imagination within a single sequence. For each predicted state, we examine the jumping flag: If $\hat{j}_t = 1$, we initiate a new imagination sequence starting from the post-jump state, which is predicted by the long-term transition model and the dynamics predictor. In other words, whenever a long-term state jump occurs, the world model generates a new sequence from the post-jump state, while the intermediate state transitions within the sequence are governed exclusively by short-term dynamics. Importantly, we optimize the actor independently for each sequence, ensuring that there is no gradient or value transfer between sequences. To evaluate the advantages of using sequential long short-term imagination, we conduct an experimental comparison between LS-Imagine (*series*) and LS-Imagine (*parallel*). Figure 6(b) shows that the LS-Imagine (*series*) outperforms LS-Imagine (*parallel*) by large margins. This implies

that the parallel imagination sequences are independent of one another, meaning that the sequence starting with a post-jumping state does not guide the prior-jumping transitions.

In the appendix, we further include (i) experiments on the long-horizon "Tech Tree" task, (ii) analyses of the long-term imagination frequency and corresponding state jumping intervals $\hat{\Delta}_t$ predicted by the model, and (iii) visualization of affordance maps with occluded target objects.

## 5  RELATED WORK

**Visual MBRL.**    Recently, learning control policies from images, *i.e.,* visual RL has been used widely, whereas previous RL algorithms learn policies from low-dimensional states. Existing approaches can be grouped by the use of model-free RL methods (Laskin et al., 2020; Schwarzer et al., 2021; Stooke et al., 2021; Xiao et al., 2022; Parisi et al., 2022; Yarats et al., 2022; Zheng et al., 2023) or model-based RL methods (Hafner et al., 2019; 2020; 2021; Seo et al., 2022; Pan et al., 2022; Zhang et al., 2023a; Mazzaglia et al., 2023; Micheli et al., 2023; Zhang et al., 2023b; Ying et al., 2023; Seo et al., 2023; Alonso et al., 2024; Hansen et al., 2024; Wang et al., 2024b). The following methods specifically enhance the modeling of long-term dynamics in visual MBRL. Lee et al. (2024b) proposed the prediction of temporally smoothed rewards to address long-horizon sparse-reward tasks. R2I (Samsami et al., 2024) improves long-term memory and long-horizon credit assignment in MBRL. Unlike existing methods, our work presents a long short-term world model architecture specifically designed for visual control in open-world environments.

**Affordance maps for robot learning.**    Our work is also related to the affordance map for robot learning (Mo et al., 2021; Jiang et al., 2021; Yarats et al., 2021; Mo et al., 2022; Geng et al., 2022; Xu et al., 2022a; Wang et al., 2022; Wu et al., 2022; Ha & Song, 2022; Xu et al., 2022b; Cheng et al., 2024; Lee et al., 2024a; Li et al., 2024b). Where2Explore (Ning et al., 2023) introduces a cross-category few-shot affordance learning framework that leverages the similarities in geometries across different categories. DualAfford (Zhao et al., 2023) learns collaborative actionable affordance for dual-gripper manipulation tasks over various 3D shapes. VoxPoser (Huang et al., 2023) unleashes the power of large language models and vision-language models for extracting affordances and constraints of real-world manipulation tasks, which are grounded in 3D perceptual space. VRB (Bahl et al., 2023) trains a visual affordance model with videos of human interactions and deploys the model in real-world robotic tasks directly. Qi et al. (2020) adopted a spatial affordance map that is trained by interacting with the environment for navigation. However, our approach distinguishes itself by employing visual observation to generate affordance maps as guidance to mitigate the low exploration efficiency in open-world environments.

**Hierarchical methods.**    Like our approach, Director (Hafner et al., 2022) learns hierarchical behaviors in the latent space, which adopts high-level policy (*manager*) to produce latent goals to guide low-level policy (*worker*). Dr. Strategy (Hamed et al., 2024) proposes strategic dreaming with latent landmarks to learn a highway policy that enables the agent to move to a landmark in the dream. Gumbsch et al. (2024) presented a hierarchy of world models, which perform high-level and low-level prediction adaptively, and the high-level predictions depend on the low-level predictions. Our method distinguishes itself by generating affordance maps through image zoom-in to encourage the agent to explicitly execute long-term imagination in the world model.

## 6  CONCLUSIONS AND LIMITATIONS

In this paper, we presented a novel approach to overcoming the challenges of training visual reinforcement learning agents in high-dimensional open worlds. By extending the imagination horizon and leveraging a long short-term world model, our method facilitates efficient off-policy exploration across expansive state spaces. The incorporation of goal-conditioned jumpy state transitions and affordance maps allows agents to better grasp long-term value, enhancing their decision-making abilities. Our results demonstrate substantial improvements over existing state-of-the-art techniques in MineDojo, highlighting the potential of our approach for open-world reinforcement learning and inspiring future research in this domain.

A limitation of LS-Imagine is the computational overhead it introduces. Additionally, its effectiveness has only been validated in 3D navigation environments with embodied agents. We aim to enhance the generalization of our approach across a wider range of tasks.

## ETHICS STATEMENT

In this work, we are committed to upholding ethical research practices. This work does not involve human subjects, personal data, or sensitive information. All environments and datasets used are synthetic and publicly available. We recognize the potential for reinforcement learning models to be misused, particularly in decision-making scenarios where unintended outcomes could arise. To mitigate these risks, we emphasize responsible deployment and encourage careful consideration of the broader impact of such systems, restricting the use of our work strictly to research purposes.

## REPRODUCIBILITY STATEMENT

We prioritize the reproducibility of our work. All results can be reproduced on publicly available RL environments by following the experimental details presented in Sec. 4 and Appendix D.6. We provide the source code at https://github.com/qiwang067/LS-Imagine.

## ACKNOWLEDGMENTS

This work was supported by the National Natural Science Foundation of China (Grants 62250062, 62302246), the Smart Grid National Science and Technology Major Project (Grant 2024ZD0801200), the Shanghai Municipal Science and Technology Major Project (Grant 2021SHZDZX0102), the Fundamental Research Funds for the Central Universities, and the CCF-Tencent Rhino-Bird Open Research Fund. Additional support was provided by the Natural Science Foundation of Zhejiang Province, China (Grant LQ23F010008), the High Performance Computing Center at Eastern Institute of Technology, Ningbo, and Ningbo Institute of Digital Twin.

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

APPENDIX

## A  ENVIRONMENT DETAILS

As illustrated in Table 2, *language description* is employed for calculating the MineCLIP reward (Fan et al., 2022). *Initial tools* are the items provided in the inventory at the beginning of each episode. *Initial mobs and distance* specifies the types of mobs present at the start of each episode and their initial distance from the agent. *Max steps* refers to the maximum allowed steps per episode.

Table 2: Details of the MineDojo tasks.

| Task | Language description | Initial tools | Initial mobs and distance | Max steps |
|---|---|---|---|---|
| Harvest log in plains | "Cut a tree." | – | – | 1000 |
| Harvest water with bucket | "Obtain water." | bucket | – | 1000 |
| Harvest sand | "Obtain sand." | – | – | 1000 |
| Shear sheep | "Obtain wool." | shear | sheep, 15 | 1000 |
| Mine iron ore | "Mine iron ore." | stone pickaxe | – | 2000 |

## B  COMPARED METHODS

We compare LS-Imagine with strong Minecraft agents, including:

- *DreamerV3* (Hafner et al., 2023): An MBRL approach that learns directly from the step-by-step imaginations of future latent states generated by the world model.

- *VPT* (Baker et al., 2022): A foundation model designed for Minecraft trained through behavior cloning, on a dataset consisting of 70,000 hours of game playing collected from the Internet.

- *STEVE-1* (Lifshitz et al., 2023): An instruction-following Minecraft agent that translates language instructions into specific goals. To evaluate its effectiveness, we assess Steve-1's zero-shot performance on our tasks by supplying it with task instructions.

- *Director* (Hafner et al., 2022): An agent learns hierarchical behaviors by leveraging a world model to plan within its latent space.

- *PTGM* (Yuan et al., 2024): An RL method that pretrains goal-based policy and adopts temporal abstractions and behavior regularization.

## C  MODEL DETAILS

### C.1  ENVIRONMENTAL INTERACTION AND DATA COLLECTION

To train LS-Imagine's world model, we collect both short-term and long-term transition data through interactions with the environment. As shown in Figure 8, at each time step $t$, the agent interacts with the environment following the current policy. At each time step, the data buffer collects a tuple $\mathcal{D}_t$, which includes $(o_t, a_t, \mathcal{M}_t, r_t, c_t, j_t, \Delta_t, G_t)$:

- $o_t$ represents the observed image.

- $a_t$ represents the agent's action taken given $o_t$.

- $\mathcal{M}_t$ is the affordance map generated by a multimodal U-Net given $o_t$ and task instructions $I$.

- $r_t$ is defined in Sec. 3.2.3, which is the immediate reward computed as a weighted sum of the sparse environmental reward $r_t^{\text{env}}$ after executing $a_{t-1}$, the MineCLIP reward $r_t^{\text{MineCLIP}}$ from a pretrained scoring model (Fan et al., 2022), and the intrinsic reward $r_t^{\text{intr}}$ defined in Eq. (1) and based on $\mathcal{M}_t$.

- $c_t$ is the continuation flag received from the environment, which indicates whether further interaction is required after this step.

- $j_t$ is the jumping flag, which is used to train the world model to trigger long-term imagination during model-based behavior learning. We first estimate the jumping probability $P_{\text{jump}}$ using Eq. (3) based on $\mathcal{M}_t$. To stabilize training, we establish a dynamic threshold $P_{\text{thresh}}$, which accounts for the varying guidance strength provided by the affordance map across different tasks, resulting in

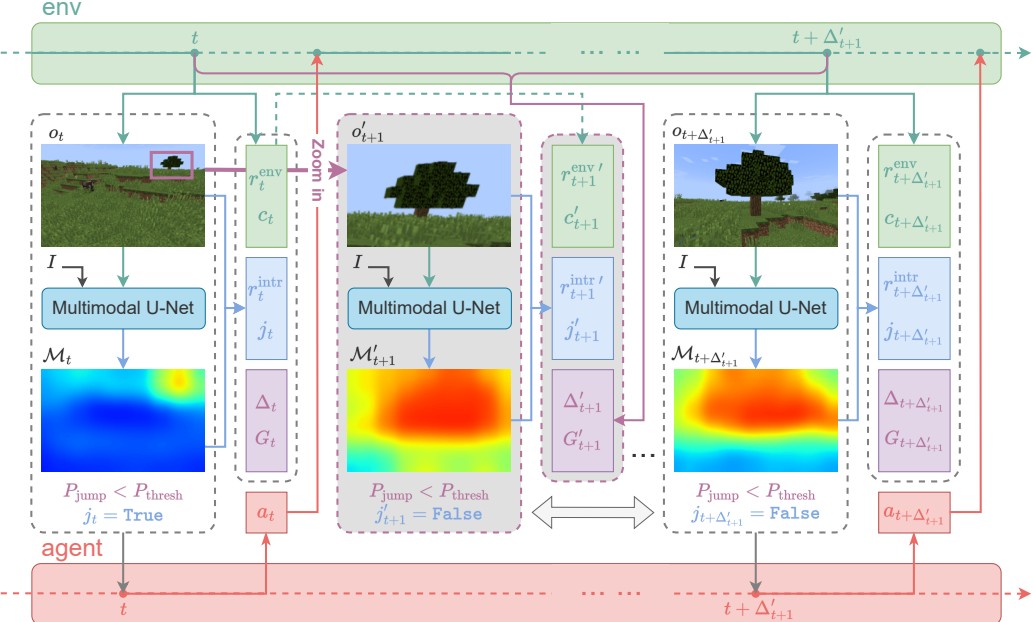

Figure 8: Environmental interaction and data collection.

task-specific distributions of $P_{\text{jump}}$. Specifically, from the beginning of training, we store the $P_{\text{jump}}$ values for every interaction step in a dedicated buffer. The threshold $P_{\text{thresh}}$ is then dynamically calculated as the mean of all $P_{\text{jump}}$ values currently in the buffer plus their standard deviation. This dynamic adjustment ensures that the threshold adapts to the characteristics of the task and remains robust throughout training. If $P_{\text{jump}} > P_{\text{thresh}}$, we set $j_t = 1$; otherwise, $j_t = 0$.

- $\Delta_t$ represents the expected number of step intervals in the jumpy state transitions during long-term imaginations. Specifically, we set $\Delta_t = 1$ by default, corresponding to a short-term transition.

- $G_t$ represents the expected cumulative reward between the pre- and post-jump states when long-term imagination occurs. Specifically, for a short-term transition, we set $G_t = r_t$ by default.

If $j_t = 0$, $\mathcal{D}_t$ is defined as the starting point of a *short*-term transition within the pair $(\mathcal{D}_t, \mathcal{D}_{t+1})$. During world model training, $(\mathcal{D}_t, \mathcal{D}_{t+1})$ is replayed to train the related modules associated with short-term dynamics. Once we obtain $j_t = 1$ during interactions, we define the current step as the starting point of a simulated *long*-term transition $(\mathcal{D}_t, \mathcal{D}'_{t+1})$. Notably, we use $\mathcal{D}'_{t+1}$ to differentiate from its short-term counterparts.

We define $\mathcal{D}'_{t+1} = (o'_{t+1}, a'_{t+1}, \mathcal{M}'_{t+1}, r'_{t+1}, c'_{t+1}, j'_{t+1}, \Delta'_{t+1}, G'_{t+1})$, where $r'_{t+1}$ and $c'_{t+1}$ are computed in the same manner as in short-term tuples but with $o'_{t+1}$ and $\mathcal{M}'_{t+1}$ as inputs. Similarly, $a'_{t+1}$ and $j'_{t+1}$ are also computed in the same way as in short-term tuples. We record them in the data buffer for better training of the reward predictor and the jump predictor.

The next question is how to annotate $\Delta'_{t+1}$, $G'_{t+1}$, and $o'_{t+1}$ to train the long-term branch.

- $o'_{t+1}$ is a simulated image rather than a real-captured image. It is obtained by cropping the original observation $o_t$ based on the high-value regions in the affordance map $\mathcal{M}_t$.

- $\Delta'_{t+1}$ is an estimation of the number of real interaction steps between the *pre-jump* state and the *post-jump* state. Since the post-jump state is not real data obtained from the environment, we first identify a real state that closely resembles the post-jump state. We then calculate the number of steps required to transition from the pre-jump state to this identified real post-jump state. Specifically, we use the intrinsic reward as a measurement. Starting from the pre-jump state, during subsequent interactions with the environment, if the agent reaches a real state where the intrinsic reward satisfies $r^{\text{intr}}_{t+\Delta'_{t+1}} \geq r^{\text{intr}}_{t+1}{}'$, we take this state as the real post-jump state and take $\Delta'_{t+1}$ as the long-term jumping interval.

- $G'_{t+1}$ is the cumulative reward within $\Delta'_{t+1}$ interaction steps, *i.e.*, $G'_{t+1} = \sum_{i=1}^{\Delta'_{t+1}} \gamma^{i-1} r_{t+i}$.

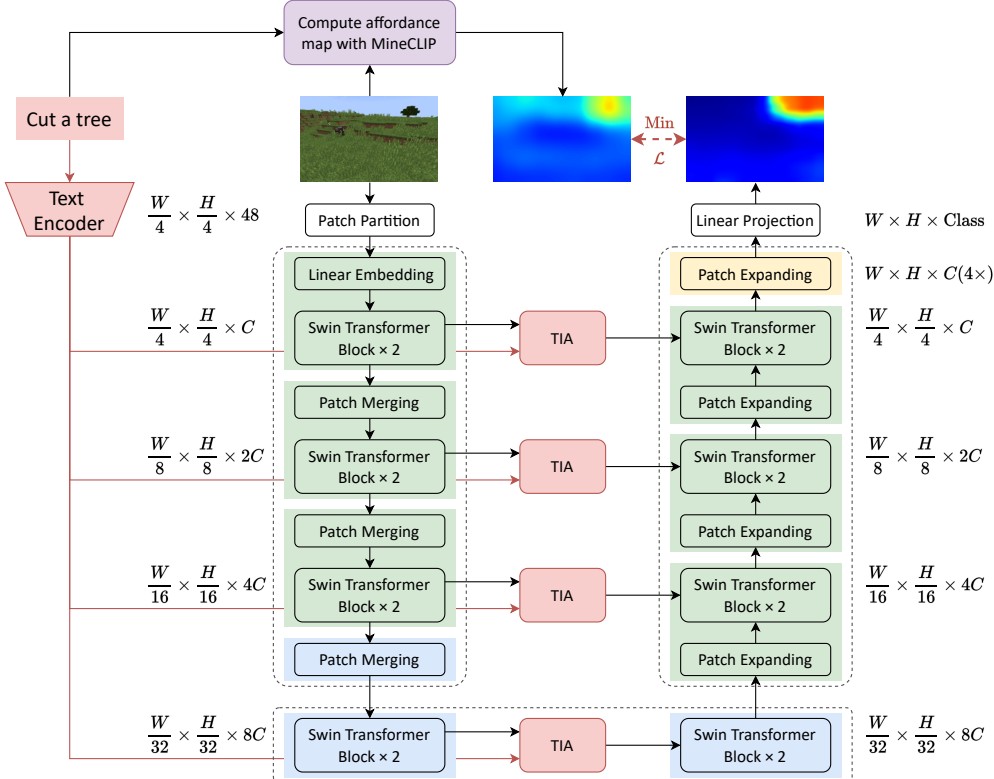

Figure 9: The architecture of multimodal U-Net.

## C.2 FRAMEWORK OF MULTIMODAL U-NET

As described in Sec. 3.2.2, we train a multimodal U-Net to rapidly generate affordance maps based on observation images and task instructions. Our enhanced multimodal U-Net architecture, as illustrated in Figure 9, is based on Swin-Unet (Cao et al., 2022), a U-shaped encoder-decoder architecture built on Swin Transformer blocks. The enhanced multimodal U-Net consists of an encoder, a decoder, a bridge layer, and a text processing module. In the Swin-Unet-inspired structure, the basic unit is the Swin Transformer block. For the encoder, the input image is divided into non-overlapping patches of size $4 \times 4$ to convert the input into a sequence of patch embeddings. Through this method, each patch has a feature dimension of $4 \times 4 \times 3 = 48$. The patch embeddings are then projected through a linear embedding layer (denoted as $C$), and the transformed patch tokens are passed through several Swin Transformer blocks and patch merging layers to produce hierarchical feature representations. The patch merging layers are responsible for downsampling and increasing the dimensionality, while the Swin Transformer blocks handle feature representation learning.

For the task instruction, the text description is processed through the text encoder of MineCLIP (Cao et al., 2022) to obtain text embeddings, which are integrated with the image features extracted at each layer of the encoder via the Text-Image Attention (TIA) module. The TIA module employs a multi-head attention mechanism to fuse image features (as keys and values) with text features (as queries) in a multi-scale attention-based fusion. The resulting fused text-image features are passed through the bridge layer and are subsequently combined with the corresponding features during the upsampling process in the decoder.

The decoder comprises Swin Transformer blocks and patch-expanding layers. The extracted context features are combined through the bridge layer with the multi-scale text-image features from the encoder to compensate for the spatial information lost during downsampling and to integrate the text information. Unlike the patch merging layers, the patch expanding layers are specifically designed for upsampling. They reshape the adjacent feature maps by performing a $2\times$ upsampling of the resolution, expanding the feature maps into larger ones. Finally, a final patch expanding layer performs a $4\times$ upsampling to restore the resolution of the feature map to the input resolution $W \times H$ ), followed by a linear projection layer applied on the upsampled features to produce pixel-level affordance maps.

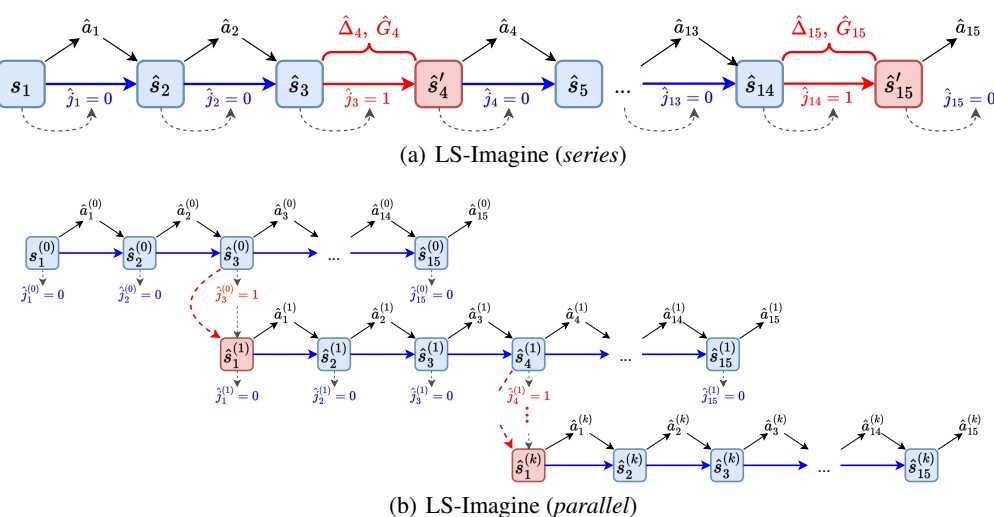

(a) LS-Imagine (*series*)

(b) LS-Imagine (*parallel*)

Figure 10: Comparison with series and parallel variants of mixed imaginations.

## C.3 VARIANTS OF LONG SHORT-TERM IMAGINATIONS

We compare two alternative pathways of the long short-term imaginations in Figure 10.

## C.4 FULL ALGORITHM

We present the training pipeline of LS-Imagine in Algorithm 1.

---

**Algorithm 1** The training pipeline of LS-Imagine.

---

1: **Initialize** parameters $\phi, \theta, \psi$.
2: Compute affordance map with MineCLIP. ▷ Affordance map generation
3: Train multimodal U-Net with annotated data. ▷ To enable real-time interaction with the affordance maps
4: Train the random agent and collect a replay buffer $\mathcal{B}$.
5: **while** not converged **do**
6:     Sample long short-term transitions from $\mathcal{B}$. ▷ Representation learning
7:     Update the world model $\phi$ using Eq. (7).
8:     Generate $(s_1, \hat{a}_1, \hat{j}_1)$ using $\pi_\theta$ and $\mathcal{W}_\phi$.
9:     **for** time step $t = 2 \cdots L$ **do** ▷ Behavior learning
10:         **if** jump flag $\hat{j}_{t-1}$ **then**
11:             Generate $(s'_t, a'_t, c'_t, j'_t, \Delta'_t, G'_t)$ using $\pi_\theta$ and long-term imagination of $\mathcal{W}_\phi$.
12:             Update $(\hat{s}_t, \hat{a}_t, \hat{c}_t, \hat{j}_t, \hat{\Delta}_t, \hat{G}_t) \leftarrow (s'_t, a'_t, c'_t, j'_t, \Delta'_t, G'_t)$.
13:         **else**
14:             Generate $(\tilde{s}_t, \tilde{a}_t, \tilde{r}_t, \tilde{c}_t, \tilde{j}_t)$ using $\pi_\theta$ and short-term imagination of $\mathcal{W}_\phi$.
15:             Update $(\hat{s}_t, \hat{a}_t, \hat{c}_t, \hat{j}_t, \hat{\Delta}_t, \hat{G}_t) \leftarrow (\tilde{s}_t, \tilde{a}_t, \tilde{c}_t, \tilde{j}_t, 1, \tilde{r}_t)$.
16:         **end if**
17:     **end for**
18:     Calculate value estimate using Eq. (9).
19:     Optimize actor $\pi_\theta$ using Eq. (11) over $\{(\hat{s}_t, \hat{a}_t, \hat{c}_t, \hat{j}_t, \hat{\Delta}_t, \hat{G}_t)\}_{t=1}^L$.
20:     Optimize critic $v_\psi$ using Eq. (10) over $\{(\hat{s}_t, \hat{a}_t, \hat{c}_t, \hat{j}_t, \hat{\Delta}_t, \hat{G}_t)\}_{t=1}^L$.
21:     **for** time step $t = 1 \cdots T$ **do** ▷ Environment interaction
22:         Sample $\hat{a}_t \sim \pi_\theta(\hat{a}_t \mid \hat{s}_t)$
23:         $r_t^{\text{env}}, o_{t+1}, c_t \leftarrow \texttt{env.step}(\hat{a}_t)$
24:         Generate affordance map $\mathcal{M}_t$ with multimodal U-Net for each $o_t$.
25:         Calculate intrinsic reward $r_t^{\text{intr}}$ and jump flag $j_t$ based on the affordance map.
26:         Collect short-term data $(o_t, a_t, \mathcal{M}_t, r_t, c_t, j_t, \Delta_t, G_t)$.
27:         **if** jumpy flag $j_t$ **then**
28:             Construct long-term data $(o'_{t+1}, a'_{t+1}, \mathcal{M}'_{t+1}, r'_{t+1}, c'_{t+1}, j'_{t+1}, \Delta'_{t+1}, G'_{t+1})$.
29:         **end if**
30:     **end for**
31:     Append long short-term transitions to $\mathcal{B}$.
32: **end while**

---

Table 3: The success rate and the number of steps per episode for task completion.

| Model | Harvest log in plains | | Harvest water with bucket | | Harvest sand | | Shear sheep | | Mine iron ore | |
|---|---|---|---|---|---|---|---|---|---|---|
| | succ. (%) | succ. step | succ. (%) | succ. step | succ. (%) | succ. step | succ. (%) | succ. step | succ. (%) | succ. step |
| **VPT** | 6.97 | 963.32 | 0.61 | 987.65 | 12.99 | 880.54 | 1.94 | 987.49 | 0.00 | — |
| **STEVE-1** | 57.00 | 752.47 | 6.00 | 989.07 | 37.00 | 770.40 | 3.00 | 992.36 | 0.00 | — |
| **PTGM** | 41.86 | 811.19 | 2.78 | 977.78 | 17.71 | 833.64 | 21.54 | 887.03 | 15.14 | **1586.03** |
| **Director** | 8.67 | 968.09 | 20.90 | 931.74 | 36.36 | 825.35 | 1.27 | 995.99 | 7.82 | 1906.31 |
| **DreamerV3** | 53.33 | 711.22 | 55.72 | 628.79 | 59.88 | **548.76** | 25.13 | 841.14 | 16.79 | 1789.06 |
| **LS-Imagine** | **80.63** | **503.35** | **77.31** | **502.61** | **62.68** | 601.18 | **54.28** | **633.78** | **20.28** | 1748.55 |

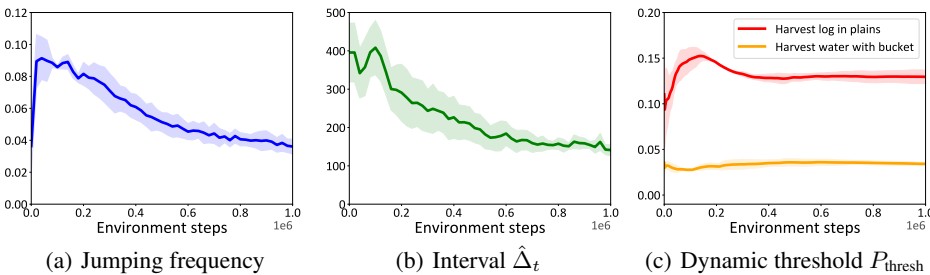

(a) Jumping frequency      (b) Interval $\hat{\Delta}_t$      (c) Dynamic threshold $P_{\text{thresh}}$

Figure 11: Analyses of long-term imaginations throughout training.

## C.5 CLARIFICATION ON STOCHASTIC LONG-TERM IMAGINATION

One might argue that long-term imagination could skip essential intermediate steps that gradually lead to the objective, potentially resulting in a lack of learning for these crucial actions. To address this issue, we adopt a probabilistic mechanism. Specifically, even when $\hat{j}_t = \texttt{True}$, indicating that a long-term transition is to be executed, we implement a probability of $0.7$ for executing the jump and $0.3$ for not jumping. This allocation ensures a $30\%$ chance that the transition will execute the short-term imagination with gradient feedback attached to the actions. This stochastic decision-making is based on a uniform distribution, providing a balanced approach between leveraging long-term imagination and capturing essential short-term behaviors.

## C.6 ADDITIONAL LIMITATION

It is worth mentioning that LS-Imagine simulates the agent's state when approaching a target object in 3D navigation environments with embodied agents by zooming in on the observed image, and sets intrinsic rewards based on whether the agent is close to and has positioned the target object at the center of the observation. Therefore, LS-Imagine is not suitable for environments with fixed viewpoints, 2D environments, or those where the reward mechanism is more complex than approaching objects (*e.g.,* driving).

## D ADDITIONAL RESULTS

## D.1 NUMERICAL COMPARISONS

Table 3 compares existing approaches on the challenging MineDojo environment.

## D.2 ANALYSES ON LONG-TERM IMAGINATIONS

We use the task *harvest log in plains* as an example to facilitate the understanding of the long short-term imagination process. In Figure 11(a), we first track the frequency of long-term imaginations and the corresponding predicted state intervals $\hat{\Delta}_t$ throughout the training process. The curve shows the proportion of imagination sequences involving jumpy state transitions relative to the total number of imagination sequences. Initially, the jumping frequency is low because the world model has not yet learned to identify when a jump is necessary based on the state. As the model's predictions improve in the early stages of training, the frequency increases, likely due to the agent's underdeveloped policies, which result in more observations far from the goal and necessitate long-term exploration. Over time, as the agent learns policies that bring it closer to the target, the frequency of observations far from the goal decreases, reducing the need for jumps.

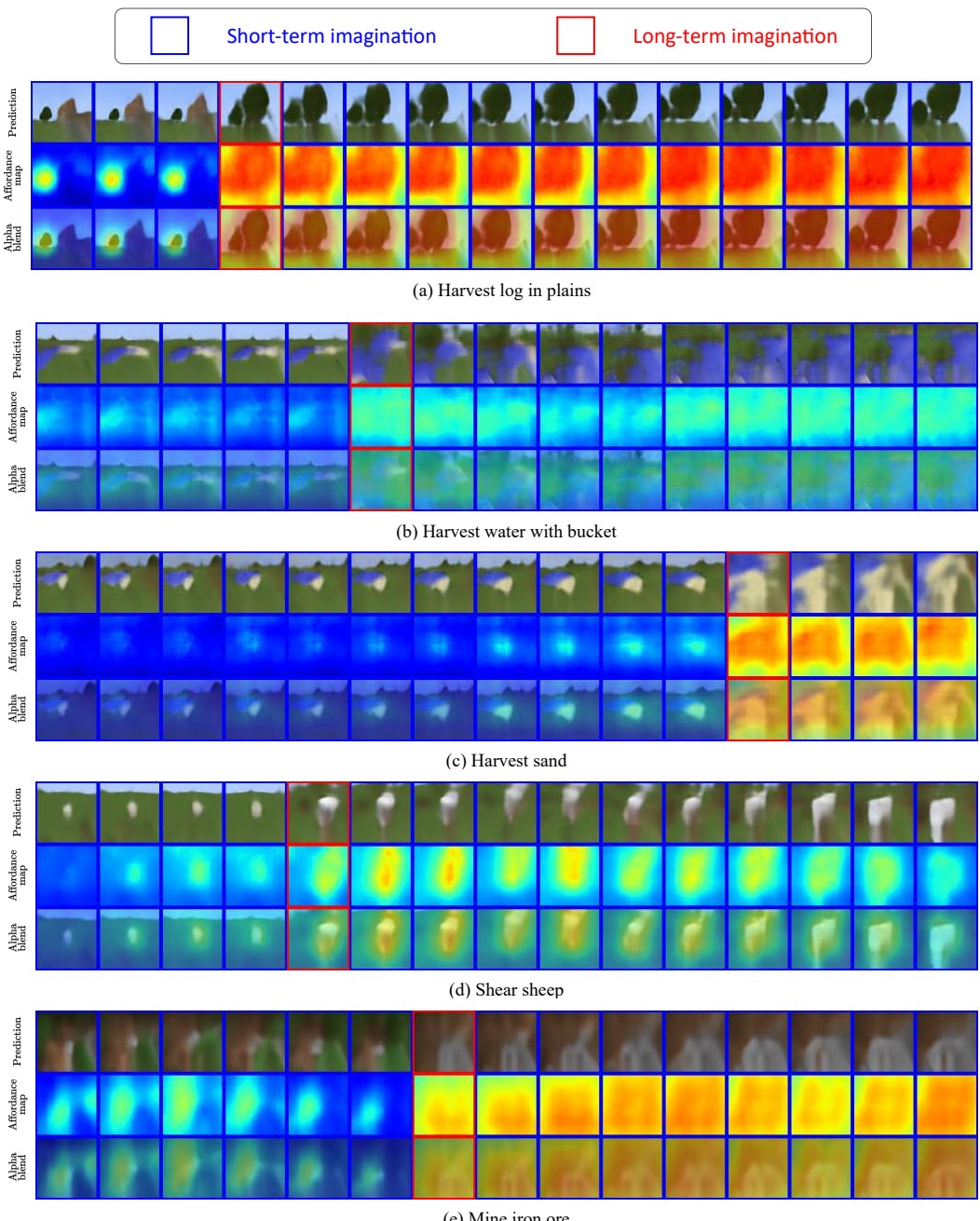

Figure 12: Visualization of the complete long short-term imagination sequences.

Additionally, we find that among all sequences with jumpy state transitions, the average number of jumpy transitions per sequence, within a horizon of $15$ steps, $1.02$. This indicates that, in most cases of these tasks, a single jumpy transition is sufficient to bring the agent close to the target.

In Figure 11(b), we track the variations of the jumping state intervals, $\hat{\Delta}_t$, throughout training. At the beginning, $\hat{\Delta}_t$ is high, indicating that the policy requires many steps to reach the target. As the policy improves, fewer steps are needed to approach the target, leading to a gradual decrease in $\hat{\Delta}_t$. Notably, as $\hat{\Delta}_t$ evolves with the updated policy, it also ensures minimal misalignment in Eq. (9) between the future cumulative rewards computed with jumpy imaginations and the behavior policy.

Furthermore, in Figure 11(c), we track the variation curves of the dynamic threshold $P_{\text{thresh}}$ during training in different tasks, and observe that:

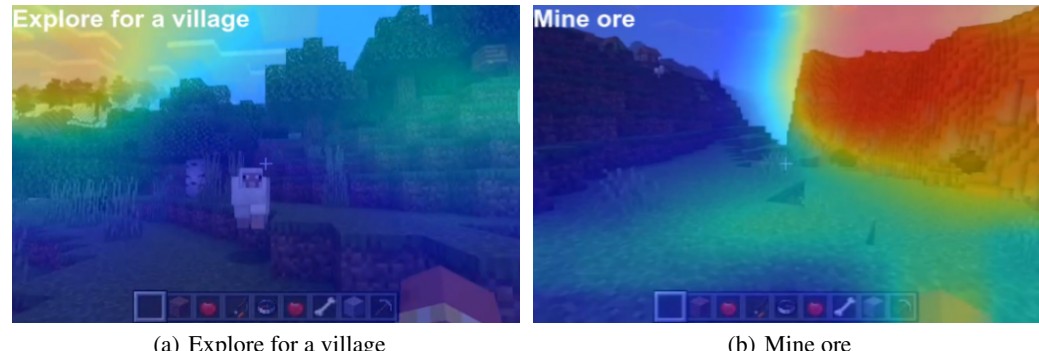

(a) Explore for a village        (b) Mine ore

Figure 13: Affordance maps when the target is invisible or occluded.

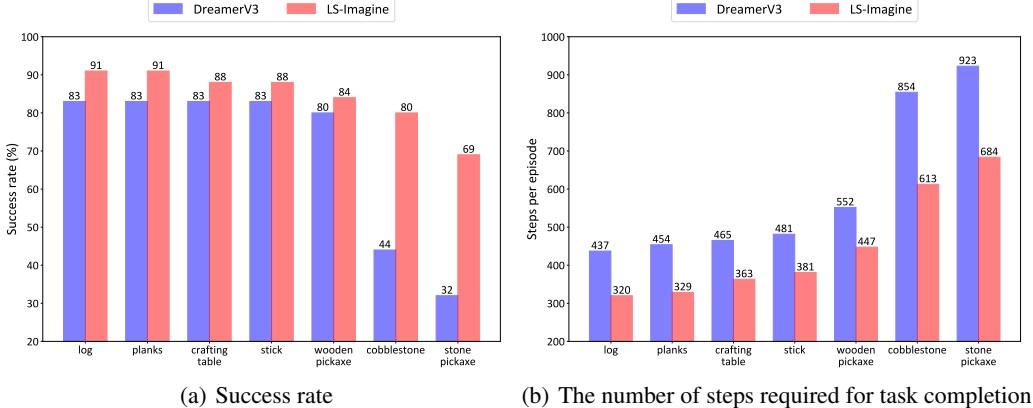

(a) Success rate        (b) The number of steps required for task completion

Figure 14: Comparison of LS-Imagine and DreamerV3 on a long-horizon "Tech Tree" task.

- For task such as *harvest log in plains*, the variance of $P_{\text{thresh}}$ is high during the early stages of training. Since $P_{\text{thresh}}$ serves as a temporal smoothing of $P_{\text{jump}}$, this reflects the significant fluctuations of $P_{\text{jump}}$ at the beginning of training, highlighting the importance of adopting a dynamic threshold.

- Across various tasks, $P_{\text{thresh}}$ consistently converges in the later stages of training, demonstrating its effectiveness in improving the stability of exploratory imaginations.

- The converged values of $P_{\text{thresh}}$ differ across tasks, indicating that involving an automated computation of $P_{\text{thresh}}$ enables us to avoid tedious hyperparameter tuning.

### D.3 VISUALIZATION OF LONG SHORT-TERM IMAGINATIONS

As illustrated in Figure 12, we visualize the complete long short-term imagination sequences for the agent across various tasks. This visualization further demonstrates how the affordance map accurately identifies regions of high exploration potential in the image, and how the long short-term imagination approach provides reasonable and applicable guidance for the agent's task execution. These qualitative results reinforce the effectiveness of our method in guiding the agent toward its goal with greater precision and efficiency.

### D.4 DEPENDENCE ON THE VISIBILITY OF OBJECTS

The long-term transitions of our approach rely on the affordance map to identify high-value exploration areas. However, it is crucial to note that our affordance map generation method is not merely an object recognition algorithm that highlights areas only when the target is present. Thanks to MineCLIP's pretraining on extensive expert demonstration videos, our approach can generate affordance maps that provide guidance even when the target is completely occluded.

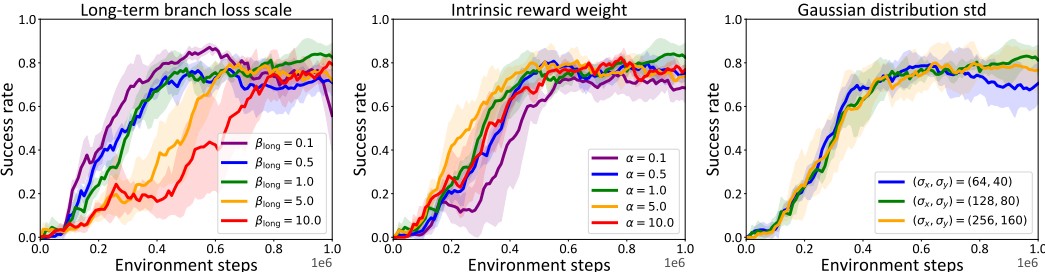

Figure 15: Results of hyperparameter sensitivity analyses.

For instance, as illustrated in Figure 13, throughout the task of locating a village, the affordance map consistently provides effective guidance to the agent, suggesting exploration of the forest to the right or the open area on the left hillside, even when the village is not visible in the current observation. Similarly, in mining tasks where ores are typically underground, the affordance map directs the agent to dig into the mountain area on the right. As we can see, even when the target is occluded, the affordance map enables the agent to continue exploring effectively.

**Further research direction.** Due to the complexity of open-world environments, the affordance map may fail to provide effective guidance in scenarios that the MineCLIP model has not encountered before. To address this issue, we plan to progressively finetune the MineCLIP model with the collected new data and introduce a new prompt to the agent: "*Explore the widest possible area to find {target}*" when the affordance map fails to identify high-value areas. This prompt, combined with intrinsic rewards generated by MineCLIP, encourages the agent to conduct extensive exploration.

### D.5    RESULTS ON LONG-HORIZON TASKS

To demonstrate the potential application of LS-Imagine in more complex tasks, we conduct experiments on a "Tech Tree" task in MineDojo, specifically *crafting a stone pickaxe from scratch*. This task involves seven subgoals: *log, planks, crafting table, stick, wooden pickaxe, cobblestone, and stone pickaxe*. Since LS-Imagine is primarily designed to focus on environmental interactions and task execution under fixed objectives, rather than task decomposition and planning, we adopt the DECKARD method (Nottingham et al., 2023) for task planning. This method provides top-level guidance, with LS-Imagine executing the corresponding subtasks. Each subtask was trained for 1 million steps and then tested within 1,000 steps per episode. The results are shown in Figure 14, which demonstrate that our LS-Imagine consistently outperforms DreamerV3, achieving higher success rates and requiring fewer steps to complete each subgoal.

### D.6    HYPERPARAMETER ANALYSES

We conduct sensitivity analyses on three hyperparameters:

- **The long-term branch loss scale** $\beta_{\text{long}}$**:** As shown in Figure 15 (Left), we observe that when $\beta_{\text{long}}$ for the long-term branch is too small or too large, it impedes the learning of long-term imagination, leading to a decline in performance.

- **The intrinsic reward weight** $\alpha$**:** From Figure 15 (Middle), we observe that if the hyperparameter $\alpha$ for intrinsic reward is excessively small, it may result in insufficient guidance and inaccurate reward estimation for the post-jumpy-transition state.

- **The intrinsic reward Gaussian parameters** $(\sigma_x, \sigma_y)$**:** As shown in Figure 16, $(\sigma_x, \sigma_y)$ control the standard deviations of the Gaussian distribution along the horizontal and vertical axes, respectively. Intuitively, setting these hyperparameters too low may cause the model to overlook targets located at the edges of the observed images. Conversely, excessively high $(\sigma_x, \sigma_y)$ may reduce the reward discrepancy for targets at different positions within the observation, thereby diminishing the agent's incentive to focus on the target precisely. From Figure 15 (Right), we observe that the final performance is robust to the tested parameters, with all configurations outperforming the baseline models presented in previous experiments.

The final hyperparameters of LS-Imagine are shown in Table 4.

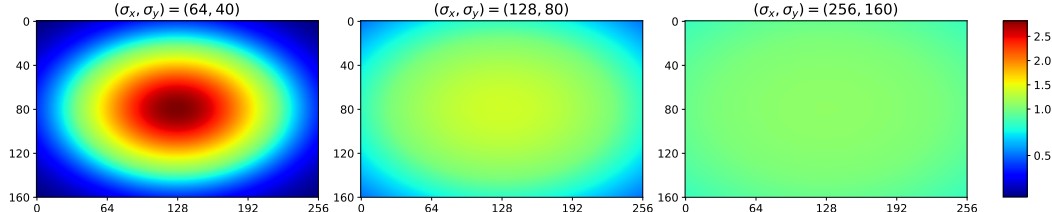

Figure 16: Visualization of Gaussian matrices with different standard deviations.

Table 4: Hyperparameters of LS-Imagine.

| Name | Notation | Value |
|---|---|---|
| **Affordance map generation** | | |
| Sliding window size | — | $0.15 \times 0.15$ |
| Sliding steps | — | $9 \times 9$ |
| U-Net train epochs | — | 500 |
| U-Net initial learning rate | — | $5 \times 10^{-4}$ |
| U-Net learning rate decay epochs | — | 50 |
| U-Net learning rate decay rate | — | 0.10 |
| Text feature dimensions | — | 512 |
| Gaussian distribution standard deviations | $(\sigma_x, \sigma_y)$ | $(128, 80)$ |
| **General** | | |
| Replay capacity | — | $1 \times 10^6$ |
| Batch size | $B$ | 16 |
| Batch length | $T$ | 32 |
| Train ratio | — | 16 |
| **World Model** | | |
| Intrinsic reward weight | $\alpha$ | 1 |
| Deterministic latent dimensions | — | 4,096 |
| Stochastic latent dimensions | — | 32 |
| Discrete latent classes | — | 32 |
| RSSM number of units | — | 1,024 |
| World model learning rate | — | $1 \times 10^{-4}$ |
| Long-term branch loss scale | $\beta_{\text{long}}$ | 1 |
| Reconstruction loss scale | $\beta_{\text{pred}}$ | 1 |
| Dynamics loss scale | $\beta_{\text{dyn}}$ | 1 |
| Representation loss scale | $\beta_{\text{rep}}$ | 0.1 |
| **Behavior Learning** | | |
| Imagination horizon | $L$ | 15 |
| Discount | $\gamma$ | 0.997 |
| $\lambda$-target | $\lambda$ | 0.95 |
| Actor learning rate | — | $3 \cdot 10^{-5}$ |
| Critic learning rate | — | $3 \cdot 10^{-5}$ |

