# OpenReview forum: "Open-World Reinforcement Learning over Long Short-Term Imagination"
_ICLR.cc/2025/Conference — ICLR 2025 Oral_

### Official Review · Reviewer_22FD · 2024-10-22

**Soundness:** 3
**Presentation:** 3
**Contribution:** 3
**Rating:** 8
**Confidence:** 4

**Summary:**

This paper proposes a long short-term world model to address the limited imagination horizon of traditional world models.
The authors leverage MineCLIP to detect task goals and generate intrinsic rewards, enabling the model to jump directly to these goal states when feasible.
For policy optimization over these jump-based transitions, they train modules to predict future value, interval timesteps, and cumulative rewards.

The paper is well-organized, and its key contributions are:
(1) It introduces a method for generating target states in MineDojo (or possibly in other 3D-RPG games).
(2) It demonstrates the feasibility of training a world model with jumping transitions and optimizing policies over such transitions.

**Strengths:**

1. It introduces a method for generating target states in MineDojo (or possibly in other 3D-RPG games).
2. It demonstrates the feasibility of training a world model with jumping transitions and optimizing policies over such transitions.
3. The illustration is clear and the writing organization is good.

**Weaknesses:**

1. Since MineCLIP is an important tool for this work, I suggest the author include a brief introduction of what MineCLIP can do in section 3.2/3.2.1 or an appropriate position. This would help readers who are not familiar with research on MineDojo to understand this paper.

2. In the abstract, "We argue that the primary obstacle in open-world decision-making is improving the efficiency of off-policy exploration across an extensive state space." This seems not closely connected to the main contribution of this paper. I suggest paraphrasing it to highlight "across a long horizon" or something that is more related to the topic.

3. Though the method sounds promising for solving MineDojo tasks, it may not be a general method for all kinds of open-world games. Such as in 2D games or fixed camera control tasks.

    (a)  Before seeing the target for the first time in the training process, there won't be a reasonable goal-directed reward or jumping option, the exploration still requires extensive enumerates.

    (b) The crop and resize operation (or assumption) is only useful for 3D visual navigation tasks.

    (c) When the target is occluded, the world model still needs to perform step-by-step imagination.

If these are true, I suggest the authors include a sentence or so in the limitation section to clarify it.

**Questions:**

1. Jumping transitions

    (a) I don't get the reason and rule for this dynamics threshold on lines 248 to 250. Could you clarify this?

    (b) The jumpy transition in the world models is a novel contribution, and I think it's worth more explanation for intuitive understanding. Like how many times would the jumping transition be triggered during the training, and what is the predicted average $\Delta'_t$ in the imagination (how many short-term steps are saved)?

2. I suggest adding a numerical result (possibly in the appendix) for the results in Figures 4 and 5.

3. For the parallel imagination in section 4.3. Is the post-jumping state connected with the bootstrapping $\lambda$-return of the pre-jumping states? How is the conclusion on lines 464 to 466 be made?

4. For the model-based RL review on lines 487-489, I suggest adding another two world model papers for completeness: (1) Zhang, Weipu, et al. "STORM: Efficient stochastic transformer based world models for reinforcement learning." Advances in Neural Information Processing Systems 36 (2024). and (2) Alonso, Eloi, et al. "Diffusion for World Modeling: Visual Details Matter in Atari." arXiv preprint arXiv:2405.12399 (2024). (NeurIPS24 Spotlight)

---

> ### Author Response · Authors · 2024-11-26
> **Responses to Reviewer 22FD (Part 1)**
>
> We appreciate your great efforts in reviewing our paper and hope that the following responses can address most of your concerns.
>
> >Q1: Add an introduction of MineCLIP in Section 3.2/3.2.1.
>
> We have added a brief introduction of MineCLIP to section 3.2.1 in the revised paper.
>
> >Q2: The abstract should be revised to emphasize "across a long horizon" to better align with the paper's main contribution.
>
> The main contribution of our approach is improving the efficiency of exploration across an extensive state space in open-world decision-making, especially for tasks that demand consideration of **long-horizon payoffs**. We refine the abstract in the revised paper.
>
>
> >Q3: Dependence on the visibility of objects.
> The long-term transitions of our approach rely on the affordance map to identify high-value exploration areas. However, it is crucial to note that our affordance map generation method is not merely a target recognition algorithm that highlights areas only when the target is present. Thanks to MineCLIP's pretraining on extensive expert demonstration videos, our approach can generate value maps that provide guidance even when the target is completely occluded.
>
> For instance, as illustrated in **Figure 13** in the revised paper, throughout the task of locating a village, the affordance map consistently provides effective guidance to the agent, suggesting exploration of the forest to the right or the open area on the left hillside, even when the village is not visible in the current observation. Similarly, in mining tasks where ores are typically underground, the affordance map directs the agent to dig into the mountain area on the right. As we can see, even when the target is occluded, the affordance map enables the agent to continue exploring effectively.
>
> Furthermore, we agree that due to the complexity of open-world environments, the affordance map may fail to provide effective guidance in scenarios that the MineCLIP model has not encountered before. To address this issue, we plan to progressively finetune the MineCLIP model with the collected new data and introduce a new prompt to the agent: "*Explore the widest possible area to find {target}*" when the affordance map fails to identify high-value areas. This prompt, combined with intrinsic rewards generated by MineCLIP, encourages the agent to conduct extensive exploration.
>
>
> >Q4: Limitation of LS-Imagine.
>
> We have included the limitation of LS-Imagine in Section 6 in the revised paper.

---

> ### Author Response · Authors · 2024-11-26
> **Responses to Reviewer 22FD (Part 2)**
>
> >Q5: The dynamics threshold on lines 248-250 needs clarification, and the "jumpy transition" should be explained further, including how often it's triggered and the average number of short-term steps saved.
>
> (1) On the dynamic threshold:
>
> We design the dynamic threshold $P_\text{thresh}$ for two purposes:
> - At the beginning of the training process, we do not have an accurate estimation of $P_{\text{jump}}$. To stabilize the training process in the imagination phase, we only trigger jumpy state transitions when $P_{\text{jump}} > P_{\text{thresh}}$, which is computed as the moving average of all $P_{\text{jump}}$ collected in the experience replay buffer.
> - Manually selecting suitable $P_{\text{thresh}}$ values for different tasks is a time-consuming work, because the threshold should account for the varying guidance strength provided by the affordance map across different tasks (as shown in **Figure 12**). Therefore, we propose to compute $P_{\text{thresh}}$ automatically.
>
> Specifically, from the very beginning of training, we store the $P_\text{jump}$ values for every interaction step in a data buffer. The threshold $P_\text{thresh}$ is then dynamically calculated as the mean of all $P_\text{jump}$ values currently in the buffer plus their standard deviation. This dynamic adjustment ensures that the threshold adapts to task-specific features and remains robust throughout training.
>
> In **Figure 11\(c\)** in the revised appendix, we track the variation curves of $P_\text{thresh}$ during training in different tasks, and observe that:
> - For task such as "harvest log in plains", the variance of $P_\text{thresh}$ is high during the early stages of training. Since $P_\text{thresh}$ serves as a temporal smoothing of $P_\text{jump}$, this reflects the significant fluctuations of $P_\text{jump}$ at the beginning of training, highlighting the importance of adopting a triggering threshold.
> - Across various tasks, $P_\text{thresh}$ consistently converges in the later stages of training, demonstrating its effectiveness in improving the stability of exploratory imaginations.
> - The converged values of $P_\text{thresh}$ differ across tasks, indicating that involving an automated computation of $P_\text{thresh}$ enables us to avoid tedious hyperparameter tuning.
>
> Please refer to Page 19 in our revision for detailed results.
>
> (2) On the frequency and jumping state intervals of the "jumpy transitions":
>
> In **Figure 11\(a-b\)**, we showcase the frequency of jumpy transitions in the imagination process and the predicted $\hat{\Delta}_t$ throughout training. We have included further analyses in **Appendix D.2 (Pages 18-19)**. Please refer to our revision.
>
> >Q6: Add a numerical result for the results in Figures 4 and 5.
>
> As suggested by the reviewer, we have added numerical results in **Table 3** in the revised appendix.
>
> >Q7: Clarify if the post-jumping state is connected with the bootstrapping $\lambda$-return of the pre-jumping states and explain the basis for the conclusion on lines 464-466.
>
> (1) The connection of post-jumping state and pre-jumping states.
>
> As indicated by Equation 9, the long-term rewards from the post-jump imaginations directly affect the bootstrapping $\lambda$-return of the pre-jump states within the same sequence, thereby influencing the value estimation of those pre-jump states.
>
> For further details, please refer to our responses to Reviewer hBEP's Q3.
>
> (2) Why post-jumping state does not guide the prior-jumping transitions in parallel variant of LS-Imagine?
>
> As illustrated in **Figure 10(b)**, an alternative method is to structure long- and short-term imagination pathways in parallel. Specifically, we begin by applying short-term imagination within a single sequence. For each predicted state, we examine the jumping flag: If $\hat{j}_t = 1$, we initiate a new imagination sequence starting from the post-jump state, which is predicted by the long-term transition model and the dynamics predictor. In other words, whenever a long-term state jump occurs, the world model generates a new sequence from the post-jump state, while the intermediate state transitions within the sequence are governed exclusively by short-term dynamics. **Importantly, we optimize the actor independently for each sequence, ensuring that there is no gradient or value transfer between sequences**.
>
> We have included these discussions in **Section 4.3** in the revision.
>
> >Q8: Add another two world model papers for completeness.
>
> We have included the suggested papers in **Section 5** in the revision.

---

> > ### Comment · Reviewer_22FD · 2024-11-28
> >
> > I appreciate the authors' efforts in the rebuttal, and the revised paper now provides a clearer and more intuitive characterization of the jumpy transition and the intrinsic reward generation. Most of my concerns have been addressed, so I have decided to raise my score to 8.

---

### Official Review · Reviewer_3G6A · 2024-10-22

**Soundness:** 3
**Presentation:** 3
**Contribution:** 4
**Rating:** 8
**Confidence:** 4

**Summary:**

This paper introduces an exploration approach, LS-Imagine, to improve model-based reinforcement learning from pixels on open-world environments (with large state spaces).

The paper aims to address the short-hoirzon limitation of model-based approaches relying on repeated autoregressive single timestep prediction. This is achieved by first generating affordance maps of observations by zooming into different patches and determining if there are objects associated with the current task in those patches using a vision-language reward model (MineCLIP in this work) distilled into an affordance map predictor. This affordance map is used to generate an intrinsic reward associated with the observation. The kurtosis of the affordance map is then used to determine if the object associated with reward is near or far away (binary, based on a threshold). If the object is deemed to be near, a standard 'short-term' world model using autoregressive single timestep prediction is used to generate trajectories for an actor and critic to learn from. Alternatively, if the object deemed to be far away, a 'long-term' world model using autoregressive multi-timestep interval prediction is used (although ignoring actions). This enables the world model to generate a trajectory over a larger number of environment timesteps in the same number of autoregressive prediction steps, therefore potentially enabling the generated trajectory to get closer to the object of interest. This long-term trajectory is used to train the critic but not the actor. The combination of these models is termed a Long Short-Term World Model, giving the approach its name: LS-Imagine.

LS-Imagine is applied to 5 MineDojo tasks and outperforms the baselines in terms of success rate and per-episode steps after training. Ablations demonstrate the benefit of both affordance-based intrinsic reward and the use of long-term imagination.

In summary, the paper attempts to tackle the important problem of long horizon world modelling and provides an interesting and novel approach to address this problem in 3D environments. While it is a relatively complex method and somewhat limited to Minecraft, I believe the ideas and insights warrant acceptance.

**Strengths:**

- **Significance**
  - Long-horizon world modeling and reinforcement learning in open-world environments are important problems.
  - The proposed approach is insightful and successfully addresses these problems.
- **Originality**
  - The proposed approach involves the combination of multiple novel and inisightful components.
- **Quality**
  - Overall the quality of the paper is relatively high, with the method reasonably clearly explained and analyzed.

**Weaknesses:**

- **Clarity**
  - Some aspects of the paper are not particularly clear. The main one is the use of the word 'jumpy' throughout the paper. The meaning of this word is assumed, but is not defined in the paper or standard usage as far as I'm aware, and is relatively unscientific, so I feel it is not the right word to use. 'Multi-step' state transitions seems more appropriate. If the authors were attempting to highlight that the number of steps can vary, then 'variable-step' transitions would be better. At the very least, 'jumpy' should be properly defined at the beginning of the paper.
  - Similarly affordance maps may not be familiar to all readers and the exact meaning of this term can vary. For example, a short clarification early in the paper such as ''...affordance maps, that elucidate which parts of an observation may be associated with reward, ..." would be helpful.
  - Some other unclear aspects/minor mistakes include:
    - L326: "employs *an* actor-critic algorithm to learn behavior *from* the latent state sequences..."
    - L351: Grammar is slightly wrong and confusing, should be: "Notably, since long-term imagination does *not* involve actions, we do not optimize the the actor when long-term imagnation is adopted."
    - L354: Also worth highlighting the difference with the DreamerV3 loss. "The loss of the actor is therefore equivalent to DreamerV3, with an additional factor of $(1-\hat j_t)$ to ignore updates from long-term imagination:"
    - L361: "on the chellenging..."
    - L500: Doesn't make sense. Maybe "Our work is also related to affordance maps for robotics." is sufficient?
    - "*Learning to Move with Affordance Maps*" [1] should likely also be compared and cited here.

- **Limitations**

  - This approach has important limitations that are not mentioned. In particular, the approach is limited to embodied agents navigating a 3D environment in which there are objects associated for which reward is obtained by approaching them. Therefore the approach assumes, for example:
    - Observations are of a 3D environment
    - Actions are navigation actions of an embodied agent
    - Rewards are assoiated with identifying or moving towards objects
    - A reward model is available to identify high reward regions of observations
  - The experiments are limited to Minecraft for which these assumptions hold. This approach would likely not work as well even in Crafter [2] for example, which provides a 2D 'open-world' analogue of Minecraft, since objects do not become larger as you move towards them.
  - The approach also relies on both the long-term and short-term models being used, given only the short-term model is able to update the actor. While the thresholding of $P_{jump}$ can partially be used to address this, this is not particularly robust, and still requires some close-up objects in initial exploration for the standard one-step world model to be used, so the approach may not work as well in very sparse environments.
  - There is still significant value of the approach despite these limitations, and the paper is reasonably scoped, but they should be included in the limitations at the end of the paper, which are currently overly brief and narrow.

  **References:**

  [1] "*Learning to Move with Affordance Maps*", Qi et al., 2020

  [2] "*Benchmarking the Spectrum of Agent Capabilities*", Hafner, 2021

**Questions:**

- Why was the word 'jumpy' chosen? Could a more precise word be used instead?
- The authors say on L42 that MBRL approaches optimize short-term experiences of "typically 50 time steps". Where did 50 come from? My understanding is that MBRL methods such as DreamerV3 commonly use an even shorter horizon of $\sim 16$ timesteps.
- Could the limitations be expanded upon to cover the general application of LS-Imagine beyond Minecraft?

---

> ### Author Response · Authors · 2024-11-26
> **Responses to Reviewer 3G6A**
>
> We thank the reviewer for the constructive comments.
>
> >Q1: Clarification of "jumpy" and "affordance map".
>
> The jumpy transition enables the agent to skip intermediate states from current state and directly simulate a future state that is more relevant to the task. We have revised the introduction section to define the term "jumpy" when it first appears in the paper. Besides, we have introduced the affordance map at the end of **Page 2**.
>
> >Q2: Some other unclear aspects/minor mistakes.
>
> We have made the following changes in the revised paper:
> *  **Line 326**: We have revised the sentence.
> *  **Line 351**: We have corrected the grammar.
> *  **Line 354**: We have highlighted the difference from the DreamerV3 loss.
> *  **Line 361**: We have checked the line and confirm that there is no typo; we originally wrote "challenging".
> *  **Line 500**: We have simplified the phrasing.
> *  We have compared and cited "Learning to Move with Affordance Maps".
>
> >Q3: Effectiveness of LS-Imagine in sparse environments.
>
> The long-term transitions in our approach leverage the affordance map to identify high-value exploration areas. Even in environments where targets are sparse, our approach continues to benefit from the affordance maps. Unlike object recognition algorithms, which highlight areas only when targets are present, our method remains effective even in the absence of clear target cues. Thanks to MineCLIP's pretraining on extensive expert demonstration videos, our approach can generate affordance maps that provide guidance even when the target is sparse or even completely occluded.
>
> For instance, as illustrated in **Figure 13** in the revised paper, throughout the task of locating a village, the affordance map consistently provides effective guidance to the agent, suggesting exploration of the forest to the right or the open area on the left hillside, even when the village is not visible in the current observation. Similarly, in mining tasks where ores are typically underground, the affordance map directs the agent to dig into the mountain area on the right. As we can see, even when the target is occluded, the affordance map enables the agent to continue exploring effectively.
>
> **Potential research direction:** Nevertheless, we agree that due to the complexity of open-world environments, the affordance map may fail to provide effective guidance in scenarios that the MineCLIP model has not encountered before. To address this issue, we plan to progressively finetune the MineCLIP model with the collected new data and introduce a new prompt to the agent: "*Explore the widest possible area to find {target}*" when the affordance map fails to identify high-value areas. This prompt, combined with intrinsic rewards generated by MineCLIP, encourages the agent to conduct extensive exploration.
>
>
> >Q4: Limitations of LS-Imagine.
>
> We have included the discussion about the limitations of our approach in **Section 6**.
>
> >Q5: On the imagination horizon of LS-Imagine.
>
> It is a typo. The imagination horizon of our approach is 15 steps. We have corrected this in the revised paper.
>
> >Q6: Could the limitations be expanded upon to cover the general application of LS-Imagine beyond Minecraft?
>
> Apart from open-world environments like Minecraft, our method is also applicable to navigation environments such as MiniWorld [1], Habitat [2], and DeepMind Lab [3]. As long as embodied agents are navigating a 3D environment where rewards are obtained by approaching objects associated with specific goals, our method can be effectively utilized.
>
>
> References:
>
> [1] Chevalier-Boisvert et al. "Minigrid & miniworld: Modular & customizable reinforcement learning environments for goal-oriented tasks", NeurIPS Track on Datasets and Benchmarks, 2024.
>
> [2] Savva et al. "Habitat: A platform for embodied ai research", ICCV, 2019.
>
> [3] Beattie et al. "Deepmind lab" arXiv preprint arXiv:1612.03801, 2016.

---

> > ### Comment · Reviewer_3G6A · 2024-11-28
> >
> > Thank you to the authors for their response and updates to the paper. Regarding the points addressed:
> >
> > $1$. The added footnote definitions of jumpy and affordance maps now make the meanings of these terms clear.
> >
> > $2$. Thank you for addressing these minor points. For line 361 (now line 405), the typo is that it should be "on *the* challenging...", not the spelling of "challenging".
> >
> > $5$. Ok, thanks for making the correction to the imagination horizon.
> >
> > $3, 4, 6$. Thank you for the explanation. The additions to Appendix D (including Figure 13) are interesting. However, I did not explicitly list sparse reward environments as a limitation. I said that the approach was limited to embodied agents navigating in 3D environments, where rewards are obtained by approaching objects associated with specific goals, as you mention in your response to Q6. Indeed the approach could also apply to the other navigation environments you mention, given a suitable reward model, but still could not be easily applied to fixed viewpoint or 2D environments, or environments where the reward is more complex than approaching objects (e.g. to name one example, driving). This is still not really explained in the newly added limitations paragraph.
> >
> > Separately, my concern with more dense/sparse environments was the sensitivity of the $P_{jump}$ parameter to different task rewards, given only the short-term model is able to update the actor. However, I see this point is now better addressed in the updated Appendix C. This better explains the thresholding procedure, which is appreciated, but still appears quite ad hoc and does not seem particularly robust in general.
> >
> > In summary, I believe the idea is an important, novel and interesting contribution for model-based approaches in embodied 3D environments, but still feel the method is somewhat complex and may not be particularly robust as a result. Additionally, I believe its application to environments outside of MineCraft may be more limited than the paper makes out. Nonetheless, I feel the paper is deserving of acceptance. As a result, I will maintain my score.

---

> > > ### Author Response · Authors · 2024-11-28
> > >
> > > >Q1: The typo in line 361 (now line 405).
> > >
> > > We will correct the typo in the revised paper.
> > >
> > > >Q2:  LS-Imagine could not be easily applied to fixed viewpoint or 2D environments, or environments where the reward is more complex than approaching objects (e.g. to name one example, driving).
> > >
> > > We will include the limitation in the revised paper.

---

### Official Review · Reviewer_ynNo · 2024-10-31

**Soundness:** 3
**Presentation:** 3
**Contribution:** 3
**Rating:** 8
**Confidence:** 4

**Summary:**

This paper studies Model-based RL in Open-World environment, specifically Minecraft. The authors propose a long short-term imagination method to improve the imagination process of the dreamer algorithm. The main idea is to use an affordance map to identify the distance object of interest and train a long-term transition model to skip the approaching steps so that the imagination can be more task-relevant. Results from 5 tasks in Minecraft showcase the improvement of the method over Dreamerv3 and other two baselines.

**Strengths:**

- The paper is mostly well written.
- The method proposed is novel and the results are promising comparing to the baselines.

**Weaknesses:**

- Although the high-level idea is straight-forward, the implementation is overcomplicated.
- The method feels very ad-hoc to the Minecraft tasks studied in this paper. It doesn't come into my mind about any other relevant tasks other than Minecraft where the proposed method can be applied.

**Questions:**

- The term "Gaussian matrix" in line 215 is confusing. Based on the context later, I think you are referring to Gaussian kernel. If that is correct, what is the $\sigma$ you use in the paper? Is the method sensitive to different value of $\sigma$?
- In Section 3.2.2, you mentioned the affordance map generator is pretrained on the random exploration data. How do you make sure that the task relevant observations are present in the random data to train a meaningful generator? For example, if the task is to collect a diamond, and there certainly won't be any diamond seen by the random agent. Will the method still help to solve the task?
- I am confused by the model analysis part with the parallel pathway. Things explained in Section 4.3 sound the same with Section 3.4. Could you elaborate what is the difference between the sequential pathway and the parallel pathway?
- Some relevant citations are missing: [1] studies the hierarchical world model which uses a similar strategy of doing the long-term imagination on the higher-level states; [2] studies a similar problem that short imagination horizon may not enough to cover behaviour of interest. I would suggest the author to discuss these works.
- There are no error bars for the middle and right plots in Figure 7. Are these results only based on one seed?
- What is the computation cost of this method?
- Maybe I missed some parts, but how does the policy learn the behaviour that the long-term imagination skip? For example, in the "Harvest log in plains" task, the go toward the tree is always skipped by the long-term imagination if the model learns well. Since the skip transition doesn't have gradient attached to the action, I wonder how does the policy learn to walk toward the tree.

[1] Gumbsch *et al.*, Learning Hierarchical World Models with Adaptive Temporal Abstractions from Discrete Latent Dynamics, ICLR 2024

[2] Hamed *et al.*, Dr. Strategy: Model-Based Generalist Agents with Strategic Dreaming, ICML 2024

---

> ### Author Response · Authors · 2024-11-26
> **Responses to Reviewer ynNo (Part 1)**
>
> Thank you for your valuable comments. We hope our responses below can help address your concerns on this paper.
>
> >Q1: The method feels very ad-hoc to the Minecraft tasks.
>
> LS-Imagine can be extended to the navigation environments, as long as embodied agents are navigating a 3D environment where rewards are obtained by approaching objects associated with specific goals, such as MiniWorld [1], Habitat [2], and DeepMind Lab [3].
>
>
> References:
>
> [1] Chevalier-Boisvert et al. "Minigrid & miniworld: Modular & customizable reinforcement learning environments for goal-oriented tasks", NeurIPS Track on Datasets and Benchmarks, 2024.
>
> [2] Savva et al. "Habitat: A platform for embodied ai research", ICCV, 2019.
>
> [3] Beattie et al. "Deepmind lab" arXiv preprint arXiv:1612.03801, 2016.
>
>
> >Q2: The term "Gaussian matrix" in line 215 is unclear and likely refers to a Gaussian kernel; please specify the $\sigma$ used in the paper and discuss the method's sensitivity to different $\sigma$ values.
>
> (1) Definition of "Gaussian matrix":
> The term "Gaussian matrix" mentioned in line 215 is indeed intended to refer to a Gaussian kernel. In our experiments, we use ($\sigma_x = 128,\sigma_y = 80$) as the standard deviations for the Gaussian kernel.
>
> The visualization of Gaussian matrices with different standard deviations is presented in **Figure 16**. As shown in the visualizations, intuitively, setting these hyperparameters too low may cause the model to overlook targets located at the edges of the observed images. Conversely, excessively high $(\sigma_x, \sigma_y)$ may reduce the reward discrepancy for targets at different positions within the observation, thereby diminishing the agent's incentive to focus on the target precisely.
>
> (2) Sensitivity of hypermeter $\sigma$:
> We present the sensitivity analysis of $\sigma$ in the **Figure 15 (right)** in the revised paper. We observe that the final performance is robust to the tested parameters, with all configurations outperforming the baseline models presented in previous experiments.
>
>
> >Q3: How are task-relevant observations ensured in the random exploration data used to pretrain the affordance map generator, particularly for tasks where random agents might not encounter task-related objects?
>
> In our current work, we employ MineCLIP to calculate the correlation between the images of simulating exploration and task description, which provides strong prior knowledge about the environment and helps guide the learning process by associating visual features with meaningful task-related concepts.
>
> We plan to further address this issue from two aspects:
> * First, we can incorporate human expert demonstrations, which provide critical examples that random agents may otherwise fail to encounter, particularly when interacting with objects related to specific tasks.
> * Second, we can iteratively update the UNet model, allowing it to progressively refine its understanding of affordances and become increasingly effective at recognizing task-relevant observations, even in more challenging scenarios.
>
>
> >Q4: The difference between the sequential pathway and the parallel pathway.
>
>
> In **Figure 10(a)** in the revised paper, we provide a visualization illustrating how the agent sequentially performs short-term and long-term imaginations within a single imagination trajectory.
>
> As illustrated in **Figure 10(b)**, an alternative method is to structure long- and short-term imagination pathways in parallel. Specifically, we begin by applying short-term imagination within a single sequence. For each predicted state, we examine the jumping flag: If $\hat{j}_t = 1$, we initiate a new imagination sequence starting from the post-jump state, which is predicted by the long-term transition model and the dynamics predictor. In other words, whenever a long-term state jump occurs, the world model generates a new sequence from the post-jump state, while the intermediate state transitions within the sequence are governed exclusively by short-term dynamics. **Importantly, we optimize the actor independently for each sequence, ensuring that there is no gradient or value transfer between sequences**.
>
> We have included these discussions in **Section 4.3** in the revision.
>
> >Q5: Some relevant citations are missing.
>
> We have added relevant citaions in **Section 5** and provided further discussions to compare these models.

---

> > ### Author Response · Authors · 2024-11-26
> > **Responses to Reviewer ynNo (Part 2)**
> >
> > >Q6: There are no error bars for the middle and right plots in Figure 7.
> >
> > We have included results averaged over three random seeds, along with the corresponding error bars in **Figure 15** in the revision.
> >
> >
> > >Q7: What is the computation cost of this method?
> >
> > Each run of LS-Imagine employs around 22GB of VRAM and takes approximately 1.8 days on a single RTX 4090 GPU.
> >
> >
> > >Q8: How the policy learns behaviors skipped by long-term imagination, given that skip transitions lack gradient feedback.
> >
> > Admittedly, long-term imaginations could skip essential intermediate steps that gradually lead to the objective, potentially resulting in a lack of learning for these crucial actions. To address this issue, we adopt a probabilistic mechanism. Specifically, even when $\hat{j}_t=\texttt{True}$ , indicating that a long-term transition is to be executed, we implement a probability of 0.7 for executing the jump and 0.3 for not jumping. This allocation ensures a 30% chance that the transition will execute the short-term imagination with gradient feedback attached to the actions. This stochastic decision-making is based on a uniform distribution, providing a balanced approach between leveraging long-term imagination and capturing essential short-term behaviors.
> >
> > Furthermore, considering that the imagination sequences can start from state encoded from an arbitary observation sampled from the replay buffer, there are possibilities that the initial states are intermediate states that be skipped by long-term transition in other imagination sequences. For example, in the *Harvest log in plains* task, the states that going toward the tree could be sampled as the initial states. Benefiting from initial states that be skipped, our approach enables the policy to learn the behavior that the long-term imagination skip.

---

> > > ### Comment · Reviewer_ynNo · 2024-11-28
> > >
> > > Thanks for your detailed rebuttal. I appreciate it especially the clarification on the application side to be more general on the navigation tasks. There is one follow-up question:
> > > - I appreciate Figure 10 to clarify the parallel pathway. But since for the parallel pathway, you branch out from one trajectory to many based on sampling $j^t$. I would think that make the batch size for policy-learning vary from batch to batch. In the meantime, the series pathway should always be based on a fixed batch size. How much difference do you think this can generate? Is it fair comparison between them?
> > >
> > > Besides that, I will raise my score to 8 to reflect the revision. Overall, I still think the idea is interesting, but the implementation is too complicated. I will encourage the authors to simplify the method to make it more approachable for the community.

---

> ### Author Response · Authors · 2024-11-28
>
> To limit the growth of parallel sequences, we impose a constraint: among the predicted states that $\hat{j}_t=\texttt{True}$ in a single imagination sequence, at most one state is selected for generating a parallel sequence via long-term transition. Since the imagination horizon of a short-term sequence is relatively short, states in the same short-term sequence typically transit to a similar post-jumping state after a long-term transition.
>
> Furthermore, as presented in **Figure 11(a)** of the revised manuscript, the frequency of long-term transitions is relatively low$\textemdash$approximately less than 10%. This ensures that the number of branching of new sequences is limited.
>
> Another potential cause for batch size growth could stem from generating additional parallel sequences on top of previously generated ones. However, we find that among all sequences with jumpy state transitions, the average number of jumpy transitions per sequence, within a horizon of 15 steps, is 1.02. This indicates that in most cases, a single jumpy transition is enough to enable the agent to approach the target, and subsequent long-term transitions are not needed.
>
> Overall, these findings ensure that the number of newly generated parallel sequences remains controlled, and batch size variations are not significant. Therefore, we believe the comparison between the parallel and series pathways is still fair.

---

### Official Review · Reviewer_hBEP · 2024-11-04

**Soundness:** 2
**Presentation:** 3
**Contribution:** 3
**Rating:** 8
**Confidence:** 3

**Summary:**

The paper introduces a novel hierarchical model-based reinforcement learning (MBRL) framework named Long Short-Term Imagination (LS-Imagine), designed to address the challenge of short-sightedness in open-world decision-making with high-dimensional visual inputs, such as MineDojo. In LS-Imagine, imagination horizon can be extended with a specialized world model. Specifically, a affordance maps are predicted to guide the jumpy switch. Two world models are trained to capture transitions at different temporal resolutions, short-term state transition and long-term state transition. Agent learning is conducted during imagination, a common strategy utilized in most background planning method.  The authors provided experimental results on Harvest tasks from the MineDojo benchmark, showed superior results compared with baseline methods.

**Strengths:**

1. The jumpy prediction technique within the long-term imagination framework is innovative as it departs from the fixed interval approach prevalent in previous work, offering increased flexibility in jumpy prediction
2. The paper is well-organized and clearly written.

**Weaknesses:**

1. The proposed method employs a hierarchical structure, yet the baseline comparisons are made with flat learning methods. Including comparisons with hierarchical MBRL methods like Director[1] could greatly strengthen the paper.
2. Equation 9 appears to have an inconsistency in the time indexing; should the bootstrapping term $R^\lambda_{t+1}$ be $R^\lambda_{t+\hat{\Delta}_{t+1}+1}$ ?
3. The use of  $\lambda$ -return in evaluating the policy might introduce bias since it should be evaluated with on-policy data,, but the predicted jumpy state, $\hat{z}_{t+1}$, might not aligned with the learning policy.
4. The paper focuses on Harvest tasks. Including results from other complex, long-horizon tasks, such as the Tech Tree task group from MineDojo, would better demonstrate the framework’s effectiveness.

[1]: Hafner, Danijar, et al. "Deep hierarchical planning from pixels." Advances in Neural Information Processing Systems 35 (2022): 26091-26104.

**Questions:**

1. Could the authors elaborate more on how the long-term transition data is collected? The phrase from the appendix still confusing to me, specifically, how is the reward of the state helps measure the long-term transition.

---

> ### Author Response · Authors · 2024-11-26
> **Responses to Reviewer hBEP (Part 1)**
>
> Thank you for reviewing our paper. Below, we provide detailed responses to your comments.
>
> >Q1: Comparison with Director.
>
> We compare the performance of Director and our approach. As shown below, LS-Imagine outperform Director in terms of success rate (%) on all tasks.
> | Task                      | Director | LS-Imagine |
> |:------------------------- | -------- | ---------- |
> | Harvest log in plains     | 8.67     | **80.63**  |
> | Harvest water with bucket | 20.90    | **77.31**  |
> | Harvest sand              | 36.36    | **62.68**  |
> | Shear sheep               | 1.27     | **54.28**  |
> | Mine iron ore             | 7.82     | **20.28**  |
>
> We have included the above results in **Table 3, Figure 4, and Figure 5** in the revised paper.
>
>
> >Q2: Inconsistency in time indexing in Equation 9.
>
> Thank you for your comment. We would like to clarify that the indexing in Equation 9 is indeed correct. Notably, Equation 9 is used in the model-based behavior learning process, where the index $t$ represents **positional order** of the imagination states predicted by the model, rather than the real time step during environmental interactions. For example, starting from state $\hat{s}\_t$, any subsequent state obtained via either a short-term transition or a long-term transition is indexed sequentially as $\hat{s}\_{t+1}$.
>
> Therefore, in Equation 9, the bootstrapping term is indexed as $R\_{t+1}^\lambda$, which is consistent with the sequential indexing convention we adopt for the imagination process; while we apply $\hat{\Delta}\_{t+1}$ to the discount factor to extend the $\lambda$-return formulation, enabling it to handle temporal spans beyond single imagination steps.
>
> For better illustration, we provide the sketch of a typical imagination sequence in **Figure 10(a)** in the revision, where the transition from $\hat{s}\_2$ to $\hat{s}\_3$ represents a short-term transition, while that from $\hat{s}\_3$ to $\hat{s}\_4^\prime$ is a long-term transition, and $\hat{\Delta}\_4$ estimates the time step interval between the pre-jump state ($\hat{s}\_3$) and the post-jump state ($\hat{s}\_4^\prime$) when the agent interacts with the real environment.
>
> We hope this clears up any confusion, and we appreciate your careful attention to this detail.
>
> >Q3: Evaluating the policy with $\lambda$-return may introduce bias since it relies on on-policy data, and the predicted jumpy states might not align with the learning policy.
>
> Let's take the imagination trajectory in **Figure 10(a)** in the revision as an example, where:
>
> $s\_1 \rightarrow \hat{a}\_1 \sim \pi\_\theta(s\_1) \rightarrow \hat{s}\_2 \rightarrow \hat{a}\_2\sim \pi\_\theta(\hat{s}\_2) \rightarrow \hat{s}\_3 \rightarrow$ Jump $\rightarrow \hat{s}\_4^\prime \rightarrow \hat{a}\_4\sim \pi\_\theta(\hat{s}\_4^\prime) \rightarrow \hat{s}\_5 \rightarrow \cdots$
>
> Although the long-term transition operation ($\hat{s}\_3 \rightarrow \hat{s}\_4^\prime$) is action-free, the discounted cumulative rewards in Equation 9 remain effective for the following reasons:
> - The rewards derived from the pre-jump actions $(\hat{a}\_1, \hat{a}\_2)$ and the post-jump actions $(\hat{a}\_4, \cdots)$ are all consistent with the current policy.
> - While ($\hat{s}\_3 \rightarrow \hat{s}\_4^\prime$) is action-free, the post-jump states $\hat{s}^{\prime}\_4$ is indirectly influenced by $\pi\_\theta$ through $\hat{s}\_3$. Therefore, Equation 9 can evaluate the potential long-term advantage of the pre-jump state $\hat{s}\_3$ by performing the $\lambda$-return iterations across the jumping operation.
> - In Equation 9, we mitigate the influence of a biased post-jump reward by applying $\hat{\Delta}_{4}$ to the discount factor when updating $R\_3^\lambda$ with $R\_4^\lambda$. Here, $\hat{\Delta}\_4$ estimates the number of intermediate steps in between when we use $\pi\_\theta$ to interact with the real environment.
>
> Technically, the $\lambda$-return in Equation 9 tries to strike a balance between the long-horizon evaluation of the current policy and mitigating reward bias. In open-world tasks, we argue that encouraging long-horizon exploration of the agent is more crucial, as it enables the agent to develop efficient policies for navigating diverse environments.

---

> > ### Author Response · Authors · 2024-11-26
> > **Responses to Reviewer hBEP (Part 2)**
> >
> > >Q4: Provide more results from additional complex, long-horizon tasks such as MineDojo’s Tech Tree task group.
> >
> >
> > We conduct experiments on tech tree tasks to further evaluate the effectiveness of our method, specifically *crafting a stone pickaxe from scratch*. This task involves seven subgoals: *log, planks, crafting table, stick, wooden pickaxe, cobblestone, and stone pickaxe*.
> >
> > Given that our approach is primarily designed to focus on environmental interactions and task execution under fixed objectives, rather than task decomposition and planning, we adopt the DECKARD method [1] for task planning. This method provides top-level guidance, with LS-Imagine executing the corresponding subtasks. We select DreamerV3 as the baseline model for comparison.
> >
> > Please refer to **Figure 14** in the revised manuscript for detailed results, which demonstrate a consistently better performance of our appraoch compared to DreamerV3 in success rates the number of steps to complete each subgoal.
> >
> > Reference:
> >
> > [1] Nottingham et al. "Do embodied agents dream of pixelated sheep: Embodied decision making using language guided world modelling", ICML, 2023.
> >
> >
> > >Q5: Elaborate on the collection of long-term transition data and clarify how state rewards are used to measure long-term transitions.
> >
> > We summarize the data collection/construction pipeline of long-term transition data as follows:
> > 1. At certain states where observations meet specific criteria (detailed in Appendix C.1), we set the current state from the real environment as the starting point of a long-term transition.
> > 2. Next, we generate the post-jump observation data $o_{t+1}^\prime$ by zooming-in certain areas in the observed $o_t$ given the affordance map $\mathcal{M}_t$.
> > 3. Since the post-jump state under $o_{t+1}^\prime$ is not obtained from true interactions, we need to find a state from the real environment that is similar to the post-jump state. Specifically, we use the intrinsic reward as a measurement. Starting from the pre-jump state, during subsequent interactions with the environment, if the agent reaches a real state where the intrinsic reward satisfies $r_{t+\Delta_{t+1}^\prime}^\text{intr} \geq r_{t+1}^{\text{intr} \ \prime}$, we take this state as the real post-jump state and take $\Delta_{t+1}^\prime$ as the long-term jumping interval.
> > 4. In the end, we compute the cumulative reward within $\Delta_{t+1}^\prime$ using $G_{t+1}^\prime = \sum_{i=1}^{\Delta_{t+1}^\prime} \gamma^{i-1} r_{t+i}.$
> >
> > Please refer to **Appendix C.1** for more details.

---

> ### Author Response · Authors · 2024-12-02
>
> Dear Reviewer hBEP,
>
> Thank you once again for your time in reviewing our paper. If any concerns remain, please don't hesitate to inform us.
>
> Best regards,
>
> Authors

---

> ### Comment · Reviewer_hBEP · 2024-12-03
>
> Thank you! I really appreciate the detailed explanation and additional experiment results from the author. My concerns have been addressed, and I would like to increase my score to 8.

---

### Author Response · Authors · 2024-11-26
**Revision Uploaded**

We thank all reviewers for the constructive comments and have updated our paper accordingly. Please take a moment to check out the revised manuscript, which incorporates the following key modifications:

**New Results:**
1. Added new baseline models: Director [1] and PTGM [2] (Figures 4-5), and provided numerical comparisons between LS-Imagine and the baseline models (Table 3, Appendix D.1).
2. Analyzed the frequency of long-term imaginations throughout training, along with the corresponding state jumping intervals $\hat{\Delta}\_t$ predicted by the model, and dynamic thresholds $P_{\text{thresh}}$ (Figure 11, Appendix D.2).
3. Added experiments on the long-horizon "Tech Tree" task, comparing our approach with DreamerV3 (Figure 14, Appendix D.5).
4. Conducted a sensitivity analysis and provided visualizations of the Gaussian matrix's parameters $(\sigma_x, \sigma_y)$ (Figures 15-16, Appendix D.6).

**Other Revisions:**
1. Revised the abstract to better clarify our main contributions.
2. Added the introduction of "jumpy transitions" and "affordance maps" (Section 1).
3. Further refined the problem formulation and notations (Section 2).
4. Added an introduction of MineCLIP (Section 3.2.1).
5. Provided clarification on the Gaussian matrix and its standard deviations (Section 3.2.3).
6. Revised the notation for long- and short-term tuples (Section 3.3.2).
7. Refined the explanation of behavior learning (Section 3.4).
8. Added the computational cost of our method (Section 4.1).
9. Clarified the difference between the series and parallel execution methods for long short-term imagination (Section 4.3, Appendix C.3, Figure 10).
10. Extended the discussion of related work and compared our method with other hierarchical RL models (Section 5).
11. Discussed more about the limitations of our method (Section 6).
12. Clarified the data collection process, with particular emphasis on how to construct long-term transition data ($\Delta_t^\prime$ and $G_t^\prime$ estimation) and how to define the dynamic threshold (Appendix C.1).
13. Presented details of the stochastic long-term imaginations (Appendix C.5).
14. Explained how the affordance map guides long-term transitions when the target is invisible or occluded (Appendix D.4).
15. Corrected the identified typos.

We appreciate the great efforts by the AC and reviewers. Please do not hesitate to let us know for any additional comments on the paper.

References:

[1] Hafner et al. "Deep hierarchical planning from pixels", NeurIPS, 2022.

[2] Yuan et al. "Pre-training goal-based models for sample-efficient reinforcement learning", ICLR, 2024.

---

### Meta-Review · Area_Chair_5T3R · 2024-12-15

**Metareview:**

The paper introduces LS-Image, a model-based RL method that uses hierarchical imagination to solve MineDojo tasks. The key idea is to use a short-term model for step-by-step transitions and a long-term one for multi-step transitions guided by learned affordance maps. These maps are computed using the MineCLIP reward model to identify task-specific spatial regions in the pixel space, which are then used to guide intrinsic rewards and long-horizon state predictions. Experimental results are shown on the MineDojo benchmark to demonstrate significant performance improvements over baseline methods in task success rate and efficiency with respect to steps per episode. There are thorough ablations to measure the effects of both affordance-driven intrinsic rewards and the long-term imagination mechanism.

This paper has an interesting and novel formulation for the hierarchical imagination framework - it is complicated but effective for 3D spatial problems. The experiments demonstrate SOTA performance on some MineDojo tasks, with relevant baselines (e.g., DreamerV3). There were problems with clarity, but the review process generated reasonable suggestions to improve that aspect.

The biggest limitation of this approach is the complexity arising from assumptions made in terms of problem domain and tasks. It has numerous components and is specific to 3D navigation problems. It will not generalize in its current form to 2D visual problems or tasks with more abstract rewards.

Despite this complexity and these assumptions, it is worth studying the effect of doing an object-oriented approach, especially since this is a recurring problem domain for all embodied agents. It might also give inspiration for future approaches to ablate certain assumptions and compare against this paper. The results are empirically well-grounded, and authors acknowledge these limitations. All of the other concerns raised by the reviewers were addressed by the authors during the review process.

**Additional Comments On Reviewer Discussion:**

Reviewer hBEP raised issues about lack of comparisons to approaches like Directory and policy evaluation for jumpy transitions. During the rebuttal process, authors ran new experiments and addressed these issues. Reviewer ynNo brought up the complexity of the method and questioned its generality to other domains, asking for clarifications on Gaussian parameters, affordance map pre-training and parallel pathways. The authors did addition sensitivity analysis, visualization and clarifications.

There were other revisions or clarifications requested but all seem to have been sufficiently addressed. So there was clear consensus amongst authors/reviewers to accept the paper.

---

### Decision · Program_Chairs · 2025-01-22

Accept (Oral)